# Efficient Kernelized Learning in Polyhedral Games beyond Full Information: From Colonel Blotto to Congestion Games

**Andreas Kontogiannis**[*]
NTUA & Archimedes / Athena RC

**Vasilis Pollatos**[*]
NKUA & Archimedes / Athena RC

**Gabriele Farina**
MIT

**Panayotis Mertikopoulos**
Univ. Grenoble Alpes, CNRS, Inria, Grenoble INP
LIG 38000 Grenoble, France & Archimedes / Athena RC

**Ioannis Panageas**
UC Irvine
Archimedes / Athena RC

## Abstract

We examine the problem of efficiently learning coarse correlated equilibria (CCE) in polyhedral games, that is, normal-form games with an exponentially large number of actions per player and an underlying combinatorial structure—such as the classic Colonel Blotto game or congestion games. Achieving computational efficiency in this setting requires learning algorithms whose regret and per-iteration complexity scale at most polylogarithmically with the size of the players' action sets. This challenge has recently been addressed in the full-information setting, primarily through the use of kernelization; however, in the more realistic partial information setting, the situation is much more challenging, and existing approaches result in suboptimal and impractical runtime complexity to learn CCE. We address this gap via a novel kernelization-based framework for payoff-based learning in polyhedral games, which we then apply to certain key classes of polyhedral games—namely Colonel Blotto, graphic matroid and network congestion games. In so doing, we obtain a range of computationally efficient payoff-based learning algorithms which significantly improve upon prior work in terms of the runtime for learning CCE.

## 1 Introduction

Learning dynamics for computing equilibria in games have been extensively studied over recent decades. The origins trace back to the work of Brown and Robinson in the 1950s [21; 51], who introduced and analyzed fictitious play. A major conceptual breakthrough came with Blackwell's approachability theorem [16], which laid the foundation for the field of online learning and, in particular, for the development of *no-regret learning* [23]. Several influential learning algorithms—such as multiplicative weights update (MWU) [7], follow-the-regularized-leader [54], and follow-the-perturbed-leader [37]—have been shown to satisfy the no-regret property. These algorithms typically maintain a probability distribution (commonly referred to as a "policy") over actions and update it iteratively, with per-iteration complexity that is polynomial in the number of actions.

Remarkably, no-regret algorithms can be used as a black-box in repeated games under the *full-information* setting, where each player observes the cost of all available actions, to recover well-established equilibrium concepts, such as coarse correlated equilibria (CCE). The no-regret property is of great importance for learning in games, as it guarantees that the time-average cost of any player using such an algorithm is no worse than the cost of the best fixed action in hindsight—regardless of

---

[*]Equal contribution.

39th Conference on Neural Information Processing Systems (NeurIPS 2025).

how the other players choose their actions. Consequently, if *all* players adopt no-regret algorithms, the learning dynamics converge to CCE.

**Polyhedral Games: motivation and challenges.** In this paper, we focus on the problem of learning CCE in multi-player games with combinatorial structure and large action spaces where the players simultaneously use no-regret learning dynamics for $T$ rounds. Specifically, we consider *polyhedral games* [32] (also dubbed *linear hypergraph games* [12]), a rich class of normal-form games where the actions per player are $d$-dimensional binary vectors with at most $m \leq d$ ones.

Polyhedral games capture important classes of games with large action sets, including the well-studied Colonel Blotto game [18], congestion games [52], extensive-form games [40], and dueling games [36]. For example, in *multi-player Colonel Blotto games*, each player must allocate $n$ soldiers among $k$ battlefields, where $n$ is typically much larger than $k$. In this case, using the one-hot representation (see Section 4), we have that $m = k$, $d = nk$ and $N = \binom{n+k-1}{k-1}$, with the latter being of order $n^k$. In *graphic matroid congestion games*, given a undirected graph $G(V, E)$, each player must choose a spanning tree, that is the basis of a graphic matroid of rank $V - 1$. In this case $m = V - 1$, $d = |E|$ and $N$ is of order $|E|^{|V|}$. Similarly, in *network congestion games*, each player needs to choose a path from $s \rightarrow t$, and the maximal path length is $K$. Here, $m = K$, $d = |E|$ and $N$ is of order $|E|^K$.

In all the aforementioned examples, the number of actions $N$ per player, grows exponentially with $m$ (approximately of order $d^m$), and as a result the vanilla learning methods for finding CCE become computationally inefficient since their per-iteration complexity is polynomial in $N$ and not polynomial in $d, m$. This computational challenge has recently been addressed in the full-information setting. Beaglehole et al. [12] demonstrated how to perform approximate fast sampling from the MWU distribution in specific polyhedral games (including the Colonel Blotto and graphic matroid congestion games). However, their approach is somewhat restrictive beyond approximately sampling from MWU, and thus sub-optimal in convergence rate, as it is unknown how to use their techniques to efficiently deploy the near-optimal Optimistic MWU [30] in such game settings. In contrast, Farina et al. [32] proposed an efficient general methodology to simulate the *exact* MWU (which allows to use optimism) algorithm via *kernelization*, requiring only $\Theta(d)$ kernel computations (see Section 2 for a formal definition) per iteration for any polyhedral game. In particular, the kernelization approach developed in [32] has led to state-of-the-art *runtime to find CCE*[*] in extensive-form games, as recently established in [31].

However, the applicability of kernelization to polyhedral games remains largely unexplored beyond the full-information setting—that is, in the *bandit* (and also *semi-bandit*) feedback settings. These settings are of particular interest in practice, as the full-information assumption—where the costs of all available actions are revealed after each round—is often unrealistic. In contrast, bandit feedback reflects a more practical regime in which only the cost of the selected action is observed. For example, learning under full-information feedback in network congestion games would impractically require each player to be able to observe the cost of *all* paths of the network, rather than just the cost of the path she actually chose.

In order to obtain equilibrium convergence guarantees in a bandit setting, the learning dynamics of each player must satisfy *no-realized-regret* guarantees that hold with *high probability* against *adaptive adversaries* (i.e., assuming that the other players can potentially adjust their policies based on the player's past actions)—a stringent and technically demanding requirement. This stands in contrast to the more commonly studied expected regret from the online learning literature (e.g., see [38; 24; 26]). An even more challenging, but very practical, requirement is to ensure that the learning dynamics achieve an *efficient runtime complexity to find CCE*, with minimal dependence on the game parameters $d$ and $m$, while still maintaining the no-regret property with favorable dependence on $T$ as much as possible.

Many algorithms from the bandit linear optimization literature [41] can be leveraged to learn $\varepsilon$-CCE in polyhedral games. Bartlett et al. [11] provide an algorithm with high probability guarantees that achieves a $\sqrt{T}$ regret bound, albeit requiring a prohibitive per-iteration complexity of poly$(N)$. The well-established GEOMETRICHEDGE algorithm (also known as COMBAND [24], or EXP2 [22]) originally proposed by Dani et al. [28] has been shown to achieve $T^{2/3}$ regret with high probability [53; 19]. Despite GEOMETRICHEDGE being a classical algorithm in the literature, how to efficiently

---

[*]The ***runtime*** of an algorithm for finding an equilibrium is defined as the product between the number of iterations $T$ needed to compute the equilibrium and the algorithm's per-iteration complexity.

implement it remains generally unclear, with path planning being the only setting where efficient implementations (e.g., via weight pushing [58; 62; 63]) were known prior to our work. Recently, Lee et al. [42] and Zimmert and Lattimore [65] proposed algorithms for continuous action spaces—which can be extended to polyhedral games using the same techniques as Abernethy et al. [1]—achieving a regret bound of $\mathcal{O}(md^{7/2}\sqrt{T})$ and $\mathcal{O}(md^2\sqrt{T})$, respectively. However, the above bounds combined with a per-iteration complexity[†], which suboptimally depends on $d$, result in impractical runtime complexity results for learning $\varepsilon$-CCE. In particular, the runtime of the algorithm in [42] to find $\varepsilon$-CCE scales as $d^{10}$, while that of [65] scales as $d^9$—both exhibiting impractically large dependence on the game parameters. Even more recently, the concurrent work of [46] proposed an algorithm for online shortest paths in DAGs with a near-optimal regret bound of $\mathcal{O}(K^{3/2}\sqrt{|E|T})$. However, their algorithm also comes with a polynomial yet impractical runtime complexity to find an approximate CCE stemming from ellipsoid method calls and other costly procedures.

Given the impractical runtime complexity results of the aforementioned approaches for learning CCE in polyhedral games, in this paper, we aim to address the following question:

> *Can kernelization techniques be extended beyond the full-information setting to design no-regret learning dynamics for computing CCE with state-of-the-art runtime complexity—achieving minimal dependence on the game parameters $d$ and $m$?*

**Main Contributions and Techniques.** In this paper, we answer the above question affirmatively. Due to the exponentially large (in $m$) per-player action sets in polyhedral games, designing efficient payoff-based learning algorithms involves addressing three primary challenges: (a) fast calculating the loss estimators which are used to update each player's policy, (b) fast sampling from each player's policy, and (c) ensuring that each player achieves efficient no-realized-regret guarantees, which imply efficiently learning $\varepsilon$-CCE.

To face the above challenges, we propose a *kernelization*-based framework, which allows us to efficiently implement standard loss estimators from bandit linear optimization. Specifically, in the bandit setting (Section 3.1), we propose a kernelized customization of the well-established GEOMETRICHEDGE algorithm [28] (see Algorithm 1). In contrast to the full-information setting, where the approach of [32] required the first moments of a MWU distribution, in the bandit setting, we require the *second moments* of MWU, needed to construct the unbiased combinatorial bandit estimator [28; 24]. Importantly, we show that we can efficiently calculate such second moments via only $\Theta(d^2)$ kernel computations (Theorem 3.1). In the semi-bandit setting, our approach (see Section 3.2) utilizes the implicit exploration loss estimator [48], which, we show that it is compatible with the kernels used for the first moment of MWU. In addition, we propose a *general efficient sampling scheme* (Procedure SAMPLING in Algorithm 1), based on kernelization, which only requires extra $\Theta(d)$ kernel computations.

Apart from improvements in the per-iteration complexity, our analysis provides the following no-regret results for learning in polyhedral games: In the *bandit setting*, we achieve $\widetilde{\mathcal{O}}(d^{2/3}m^{4/3}T^{2/3})$ regret with high probability (Theorem 3.2), improving upon baselines [53; 19] in the dependence on the game parameters. Moreover, we achieve better regret than [65] in the realistic regime where $T \leq d^6$. Regarding the *semi-bandit setting*, we achieve $\widetilde{\mathcal{O}}(m\sqrt{Td})$ regret with high probability (Theorem 3.4), which is a factor $\sqrt{m}$ worse than the optimal expected regret guarantee [8]. To the best of our knowledge, this is the first high probability result on the general setting.

To showcase the power of our general framework, we study three important classes of polyhedral games: the multi-player Colonel Blotto, graphic matroid and network congestion games.

In *Colonel Blotto games*, we use kernelization techniques based on the generator function induced by the game's combinatorial structure, in order to efficiently compute the required kernels. Remarkably, our kernelization-based approach operates directly on the geometry of the Colonel Blotto game, by allowing us to leverage an efficient $\Theta(nk)$-representation. Prior work had only been able to operate with DAG representations of the set, leading to suboptimal formulations with $\mathcal{O}(n^2k)$ edges. As shown in Table 1 (and stated in Theorem 4.4), in the bandit setting, our approach learns an $\varepsilon$-CCE in time $\widetilde{\mathcal{O}}\left(n^{2+\omega}k^{6+\omega}/\varepsilon^3\right)$—where $\omega$ is the multiplication exponent (currently the best known is

---

[†]The per-iteration complexity of the algorithm in [42] is $\widetilde{\mathcal{O}}(d^3)$ due to the fact that the optimization step is solved via an interior point method (see [2]), while that of [65] is $\widetilde{\mathcal{O}}(d^5)$ due to the pre-processing step needed to sample from a log-concave distribution (see [44]).

| Algorithm | Runtime to $\varepsilon$-CCE | Representation | Feedback |
|---|---|---|---|
| Beaglehole et al. [12] | $\widetilde{\mathcal{O}}(nk^4/\varepsilon^2)$ | $\mathcal{O}(k\log n)$ | Full-Info |
| **Our Work** | $\widetilde{\mathcal{O}}(\vert\mathcal{P}\vert nk^3/\varepsilon)$ | $\mathcal{O}(nk)$ | Full-Info |
| **Our Work** | $\widetilde{\mathcal{O}}\left(n^2k^4/\varepsilon^2\right)$ | $\mathcal{O}(nk)$ | Semi-Bandit |
| Leon et al. [43] | $\widetilde{\mathcal{O}}\left(\frac{n^4k^5}{\varepsilon^3}\left(\max\left\{\frac{1}{\lambda_{\min}},n^2\right\}\right)^{\frac{3}{2}}\right)^{\ddagger}$ | $\mathcal{O}(n^2k)$ | Bandit |
| Zimmert and Lattimore [65] | $\widetilde{\mathcal{O}}(n^{18}k^{11}/\varepsilon^2)^{\dagger}$ | $\mathcal{O}(n^2k)$ | Bandit |
| **Our Work** | $\widetilde{\mathcal{O}}\left(n^{2+\omega}k^{6+\omega}/\varepsilon^3\right)$ | $\mathcal{O}(nk)$ | Bandit |

Table 1: Comparison of results in ***Colonel Blotto games***, split by feedback type (full-information, semi-bandit, and bandit). †: The approach of [65] is evaluated using the layered graph polytope [61] of size $n^2k$. ‡: The runtime of [43] depends on the arbitrarily large $1/\lambda_{\min}$ – that is, the inverse of the minimum eigenvalue of $\mathbb{E}[vv^T]$ under the exploration distribution.

| Algorithm | Runtime to $\varepsilon$-CCE | Feedback |
|---|---|---|
| [12] | $\widetilde{\mathcal{O}}(\vert V\vert^5/\varepsilon^2)$ | Full-Info |
| **Our Work** | $\widetilde{\mathcal{O}}\left(\vert P\vert\vert V\vert^4(\vert V\vert^{\omega-1}+\vert E\vert)/\varepsilon\right)$ | Full-Info |
| **Our Work** | $\widetilde{\mathcal{O}}(\vert E\vert^2\vert V\vert^{2+\omega}/\varepsilon^2)$ | Semi-Bandit |
| [65] | $\widetilde{\mathcal{O}}(\vert V\vert^{29}/\varepsilon^2)$ | Bandit |
| **Our Work** | $\widetilde{\mathcal{O}}\left(\vert E\vert^3\vert V\vert^6(\vert V\vert^{\omega-1}+\vert E\vert)/\varepsilon^3\right)$ | Bandit |

Table 2: Comparison in ***Graphic Matroid Congestion Games***. To assess [65], we used the polytope representation of [47] that uses $d = \vert V\vert^3$ and has a small number of constraints.

| Algorithm | Runtime to $\varepsilon$-CCE | Feedback |
|---|---|---|
| [35] | $\widetilde{\mathcal{O}}(\vert E\vert^{1+\omega}K^3/\varepsilon^2)$ | Semi-Bandit |
| [49] | $\widetilde{\mathcal{O}}(\vert E\vert^9/\varepsilon^4)^{*}$ | Semi-Bandit |
| **Our Work** | $\widetilde{\mathcal{O}}(\vert E\vert^{1+\omega}K^2/\varepsilon^2)$ | Semi-Bandit |
| [65] | $\widetilde{\mathcal{O}}(\vert E\vert^9K^{10}/\varepsilon^2)$ | Bandit |
| **Our Work** | $\widetilde{\mathcal{O}}(\vert E\vert^{2+\omega}K^4/\varepsilon^3)$ | Bandit |

Table 3: Comparison of results in ***Network Congestion Games***. ∗: The algorithm proposed in [49] also achieves convergence to Nash equilibria, albeit with slower rates.

$\approx 2.372$ [5])—thereby significantly improving over [65] in the dependence on the game parameters by a factor $\approx n^{13}k^2$. In the semi-bandit setting, our approach learns an $\varepsilon$-CCE in time $\widetilde{\mathcal{O}}\left(n^2k^4/\varepsilon^2\right)$.

In *graphic matroid congestion games*, we design kernelization techniques based on the celebrated Matrix-Tree Theorem [59]. To reduce the amortized kernel computation time, we use fast rank-1 updates of the LU decomposition of Laplacian matrices based on the structure of the required kernels. Moreover, we perform efficient exact sampling via an incremental kernelization approach. As shown in Table 2 (and also in Theorem 5.2), in the bandit setting, our approach learns an $\varepsilon$-CCE in time $\widetilde{\mathcal{O}}\left(\vert E\vert^3\vert V\vert^6(\vert V\vert^{\omega-1}+\vert E\vert)/\varepsilon^3\right)$, significantly improving upon the very impractical dependence on $\vert V\vert^{29}$ of [65]. In the semi-bandit setting, our approach learns an $\varepsilon$-CCE in time $\widetilde{\mathcal{O}}(\vert E\vert^2\vert V\vert^{2+\omega}/\varepsilon^2)$.

**Remark 1.1.** *We can combine our kernelization results for Colonel Blotto and graphic matroid congestion games with the full-information framework developed in [32] to yield $1/\varepsilon$ convergence to $\varepsilon$-CCE in these games—thus addressing an open question of Beaglehole et al. [12]*

**Remark 1.2.** *Our kernelization techniques developed for graphic matroids (see Lemma 5.1) allow us to efficiently implement* GEOMETRICHEDGE *over spanning trees—thus, to the the best of our knowledge, resolving an open question posed by Cesa-Bianchi and Lugosi [24].*

In *network congestion games*, our framework improves upon [35; 27; 49; 65]. For a summary of our results, we refer to Table 3. Due to space constraints, our formal results can be found in Appendix I. Further details on existing approaches for each of the above games can be found in Appendix A.

## 2 Preliminaries

**Polyhedral Games.** In this paper, we consider *Polyhedral Games*, a structured class of normal-form games with exponentially large action sets, where each action can be represented as a binary

$d$-dimensional vector of at most $m$ ones and the incurred cost is linear in the action vector. For simplicity, here we assume that all players have the same action sets. Formally, we represent a polyhedral game as a tuple $\mathcal{G} = (\mathcal{P}, \mathcal{V}, \{L_i\})$. The set $\mathcal{P}$ defines the set of players, each of which is assigned a unique player identifier in $[|\mathcal{P}|] := \{1, 2, \ldots, |\mathcal{P}|\}$. The finite set $\mathcal{V} \subset \mathbb{R}^d$ of size $N$ represents the actions available to each player $i \in [|\mathcal{P}|]$, such that for any $v \in \mathcal{V}$, $\|v\|_1 \leq m$. We denote by $-i$ all agents except $i$. We define the loss vector function $\ell_i : \mathcal{V}^{|P|} \to \mathbb{R}_+^d$. $L_i : \mathcal{V}^{|P|} \to \mathbb{R}_+$ is the cost function, which is linear in $v_i$; that is, $L_i(v_i; v_{-i}) = \ell(v_i; v_{-i}) \cdot v_i$.

**Online Learning Setup in Polyhedral Games.** In polyhedral game dynamics under partial-information feedback, each player iteratively updates her strategies based on the feedback she receives about the loss. We consider the bandit and semi-bandit settings. In the semi-bandit setting, each player $i$ selects an action $v_i \in \mathcal{V}_i$ and receives the losses $\ell_i(j)$ of the loss vector $\ell_i = \ell_i(v_i; v_{-i})$ for all $j$ such that $v_i(j) = 1$. In the bandit setting, each player receives only $L_i(v_i; v_{-i})$.

However, it is not clear how a selfish player $i$ should update her strategy in order to minimize her overall loss, since the strategies of the other players can arbitrarily change over time. Thus, player $i$ tries to minimize her experienced loss under the worst-case assumption that the loss of each coordinate is selected by a malicious adversary.

Based on the above, we focus on the single-player's perspective and examine an abstract—online learning—model, where each player is a decision maker interacting with an unknown and potentially adversarial environment. At each round $t = 1, 2, \ldots, T$ of the online learning process, the decision maker samples an action $v_t \in \mathcal{V}$ from a probability distribution $p_t \in \Delta(\mathcal{V})$. Subsequently, the environment chooses a loss vector $\ell_t \in \mathbb{R}^d$, potentially in an adversarial fashion. This is the same setup adopted in [49; 27]. Given any round $T$, we define the *regret* up to round $T$ as follows:

$$R_T = \sum_{t=1}^{T} v_t \cdot \ell_t - \min_{v^* \in \mathcal{V}} \sum_{t=1}^{T} v^* \cdot \ell_t. \tag{1}$$

We note that the above notion measures *realized* regret, that is, it measures the performance of the algorithm based on the actions *sampled* from the distribution $p_t$. We say that players are playing no-regret learning in the game if each one of them achieves sublinear regret.

A prominent result in the theory of learning in games establishes a celebrated connection between no-regret learning and CCE of the game (which, in two-player zero-sum games, are Nash equilibria).

**Theorem 2.1** (Informal, [34]). *Suppose $|P|$ players are playing no-regret learning in the game. Let $\sigma^* := \frac{1}{T} \sum_{t=1}^{T} v_1^{(t)} \otimes \cdots \otimes v_{|\mathcal{P}|}^{(t)}$ be the time-average joint actions over $T$ rounds. Then, $\sigma^*$ forms an $T^{-1} \max(R_{T,1}, \ldots, R_{T,|\mathcal{P}|})$-approximate CCE of the game, where $R_{T,i}$ is the regret for the $i$-th player at the $T$-th round.*

**Kernelized MWU.** Multiplicative Weights Update (MWU) is an online learning algorithm that iteratively updates a distribution $p_t$ over actions in $\mathcal{V}$. Let $p_0 := \frac{1}{|\mathcal{V}|} \mathbf{1} \in \Delta(\mathcal{V})$. The MWU rule at each time step $t$ is $p_t(v) \propto p_{t-1}(v) \cdot e^{-\eta_t w_t(v)}, \quad \forall v \in \mathcal{V}$. In the standard MWU variant we set $w_t := \ell_{t-1}$, where $\ell_{t-1}$ is the loss vector observed at $t - 1$. By setting $w_t := 2\ell_{t-1} - \ell_{t-2}$ we derive the *Optimistic MWU* (OMWU) algorithm [30], which achieves constant regret (up to logarithmic factors), and thus $1/\varepsilon$ convergence to CCE in the context of learning in games.

In polyhedral games, we are interested in the efficient calculation of moments of the MWU distribution. For this aim, a useful tool introduced in [58; 32] is the **kernel function** $\mathbb{R}^d \times \mathbb{R}^d \to \mathbb{R}$ defined as follows: $K_{\mathcal{V}}(x, y) := \sum_{v \in \mathcal{V}} \prod_{j: \in v(j) = 1} x(j) \, y(j)$. The next theorem shows how to compute the first moment of $p_t$ via $d + 1$ kernel computations.

**Theorem 2.2** (First Moment Calculation, [32]). *At all rounds $t \geq 0$, the first moment of the MWU distribution, $p_t$, can be calculated as follows:*

$$\mathbb{E}_{v \sim p_t}[v] = \left(1 - \frac{K_{\mathcal{V}}(C_t, \bar{e}_1)}{K_{\mathcal{V}}(C_t, \mathbf{1})}, \ldots, 1 - \frac{K_{\mathcal{V}}(C_t, \bar{e}_d)}{K_{\mathcal{V}}(C_t, \mathbf{1})}\right),$$

*where $C_t(j) := \exp\left\{-\sum_{\tau=1}^{t} \eta_\tau w_\tau(j)\right\}$ and $\bar{e}_j(h) := \mathbb{1}\{h \neq j\}$, for $h, j \in [d]$.*

# 3 Kernelized Payoff-based Learning in Polyhedral Games

In this section, we design a framework for efficient payoff-based learning in polyhedral games, under bandit and semi-bandit feedback. Upon this framework, we build learning algorithms, which achieve efficient no-realized-regret learning with high probability guarantees against adaptive adversaries—a key requirement to show convergence to CCE. For our algorithms to be efficiently implementable, as we will see, it suffices that the kernels used for constructing the loss estimators and for sampling (highlighted in orange in Algorithm 1) can be computed efficiently. This can be achieved by effectively leveraging the game's combinatorial structure, as we will explain in depth later in the paper. For the remainder of this section, we assume that we have oracles for calculating the required kernels. In the next sections, we will demonstrate how our algorithms can be implemented efficiently in prominent examples of polyhedral games.

## 3.1 Kernelized GEOMETRICHEDGE for Bandit No-Regret Learning

---

**Algorithm 1:** Kernelized GEOMETRICHEGDE

---

**Data:** $d, m, \eta > 0, \gamma \in [0, 1]$

1 Compute a 2-approximate-barycentric-spanner $B$

2 Initialize $q_0 = [1/N, \ldots, 1/N] \in \Delta(\mathcal{V})$, $\mu = \frac{1}{d}\mathbb{1}\{v \in B\}$, $c_0(j) = 0$ and $C_0(j) = 1$, $\forall j \in [d]$

3 **for** $t = 1, \ldots, T$ **do**

4      Mixing:     $p_t = (1 - \gamma)q_t + \gamma\mu$,     where $q_t = \text{MWU}(C_t)$

5      Compute the kernels:     $K_{\mathcal{V}}(C_{t-1}, \mathbf{1})$ and $\{K_{\mathcal{V}}(C_{t-1}, \bar{e}_{j,j'})\}, \forall j, j' \in [d]$

6      Sample $v_t \sim (1 - \gamma)\text{SAMPLING}(\mathcal{V}, C_{t-1}) + \gamma\mu$

7      Observe the bandit loss $L_t = \ell_t \cdot v_t$

8      Compute $\Sigma_t(q_t)$ using Theorem 3.1 and set $\Sigma_t = (1 - \gamma)\Sigma_t(q_t) + \frac{\gamma}{d}BB^T$

9      Compute the unbiased loss estimator:     $\widehat{\ell}_t = L_t\Sigma_t^{-1}v_t$

10     Update the aggregated loss estimators:     $c_t(j) = c_{t-1}(j) + \widehat{\ell}_t(j), \quad \forall j \in [d]$

11     Update the exponential cumulative loss estimators:     $C_t(j) = \exp(-\eta c_t(j)), \quad \forall j \in [d]$

12

13 **Procedure:** SAMPLING

14 **Input:** $\mathcal{V}, C$

15 Sample $v[1] \sim Be\left(1 - \frac{K_{\mathcal{V}}(C, \bar{e}_1)}{K_{\mathcal{V}}(C, \mathbf{1})}\right)$

16 **for** $j = 2, \ldots, d$ **do**

17      Compute the kernel: $K_{\mathcal{V}(j)}$

18      Set $\mathcal{V}(j) = \{v' \in \mathcal{V} : v'[i] = v[i], \forall i \in [j-1]\}$ and $p_j = 1 - \frac{K_{\mathcal{V}(j)}(C, \bar{e}_j)}{K_{\mathcal{V}(j)}(C, \mathbf{1})}$

19      Sample $v[j] \sim Be(p_j)$

20 **Return:** $v$

---

We present our first algorithm, which establishes efficient no-regret learning in polyhedral games under bandit feedback. Our algorithm (Algorithm 1) is a *kernelized* customization of GEOMETRICHEDGE [28], a classical algorithm in the study of combinatorial bandits [41]. Despite GEOMETRICHEDGE being an algorithm with a well-studied expected regret analysis, how to efficiently implement it remains largely unclear. The primary challenges in applying the vanilla method are the following:

1. Calculating $\Sigma = \mathbb{E}[vv^T]$ – needed to construct the unbiased loss estimates which will be used by a MWU routine – in poly$(d, m)$ time.

2. Sampling from MWU in poly$(d, m)$ time.

In this paper, we tackle the above challenges in the context of polyhedral games (however, our approach can also be applied to the well-studied combinatorial settings discussed in [24]). The main idea of our approach is to utilize a loss estimate for each coordinate $j \in [d]$, which can be

kernelized efficiently, and simulate MWU using a fast sampling schema based on the computed kernels. Subsequently, we present the main components of our approach.

**Second Moment Kernelization.** Algorithm 1 uses a distribution $p_t$ which is the mixture between a MWU distribution $q_t$ and the uniform distribution, $\mu$, over a 2-approximate barycentric spanner of $\mathcal{V}$. Due to space constraints, we prompt the interested reader to Appendix E for background on barycentric spanners. In contrast to the full-information setting, where kernelized MWU [32] requires the first moment of the MWU distribution to simulate the update rule, our algorithm also requires the *second moment* of $p_t$ (i.e., the autocorrelation matrix, denoted by $\Sigma_t$) to construct the standard unbiased estimator of GEOMETRICHEDGE (Step 9). Through Step 8, it suffices to efficiently calculate $\Sigma_t(q_t)$, that is the autocorrelation matrix under the law of $q_t$, which in general was not known how to efficiently compute prior to our work (with the only exception being path planning problems where weight pushing techniques [58; 53] can be applied).

For this purpose, we will make use of kernelization. The next theorem shows that we can efficiently calculate the second moment of $q_t$ using only $d^2 + 1$ kernel computations [‡].

**Theorem 3.1** (Second Moment Calculation). *Let $\Sigma_t(q_t) := \sum_{v \in \mathcal{V}} q_t(v) \cdot (vv^T)$ be the autocorrelation matrix under the law of a MWU distribution $q_t$. Then, for all $j, j' \in [d]$,*

$$\Sigma_t(q_t)[j, j'] = 1 - \frac{K_{\mathcal{V}}(C_t, \bar{e}_j) + K_{\mathcal{V}}(C_t, \bar{e}_{j'}) - K_{\mathcal{V}}(C_t, \bar{e}_{j,j'})}{K_{\mathcal{V}}(C_t, \mathbf{1})}$$

*where $\bar{e}_{jj'}(h) := \mathbb{1}\{h \neq j \text{ and } h \neq j'\}$, for $h, h', j \in [d]$ and $\bar{e}_j(h) := \mathbb{1}\{h \neq j\}$, for $h, j \in [d]$.*

**Kernelization implies efficient exact sampling.** Based on kernelization, we propose an efficient sampling scheme (Procedure SAMPLING in Algorithm 1) that only requires extra $d$ kernel computations. We are interested in sampling $v \sim p_t$. By using the chain rule on the probability of the intersection events, we derive the following:

$$p_t(v) = \Pr[v(1)] \Pr[v(2)|v(1)] \cdots \Pr[v(d)|v(1), \ldots, v(d-1)] \tag{2}$$

It is easy to see that the $j$-th term of the above product equals the $j$-th coordinate of the first moment kernelization (see Theorem 2.2 and Observation 3.3) of the conditional polytope $\mathcal{V}(j)$—i.e., the polytope which has the first $j - 1$ coordinate values equal to the $j - 1$ sampled values (Step 18). Based on this, we iteratively sample each coordinate $j \in [d]$ from a Bernoulli distribution (Step 19), which has probability equal to $p_j = \Pr[v(j)|v(1), \ldots, v(j-1)] = 1 - \frac{K_{\mathcal{V}(j)}(C, \bar{e}_j)}{K_{\mathcal{V}(j)}(C, \mathbf{1})}$ (Step 19).

**Improved regret dependence on the game parameters.** Our analysis differs from that of the original paper of GEOMETRICHEDGE [28], which studied expected regret. Importantly, we improve upon prior analyses [53; 19], which also studied the realized regret of the algorithm, by reducing the regret's dependence on $d$ and $m$, while avoiding dependence on the possibly exponentially small minimum eigenvalue, $\lambda_{\min}$, of the autocorrelation matrix under the law of the initial distribution. In particular, we achieve this by using a more careful analysis on the effect of the barycentric spanner to the variance of the estimator. The following theorem shows that Algorithm 1 is no-regret.

**Theorem 3.2** (No-Regret under Bandit Feedback). *For $T \geq 8d^2m$, by setting $\gamma = \frac{d^{2/3}m^{1/3}}{T^{1/3}}$ and $\eta = \frac{1}{4d^{4/3}m^{2/3}T^{1/3}}$, Algorithm 1 achieves regret $R_T \leq \widetilde{\mathcal{O}}(d^{2/3}m^{4/3}T^{2/3})$ with high probability.*

### 3.2 The Semi-Bandit Feedback Case: Kernelizing Implicit Exploration

Now, we discuss our second learning algorithm for polyhedral games which establishes efficient no-regret learning under semi-bandit feedback. The main idea is similar to that of the bandit setting—that is, we utilize a loss estimator for each coordinate $j \in [d]$, which can be kernelized efficiently, and use the SAMPLING procedure to fast sample from a MWU routine using the computed kernels. Due to the exponentially large action set $\mathcal{V}$, one challenge here is that it is intractable to brute-force over $\mathcal{V}$ in order to compute the unconditional probabilities, $\Pr[v_t(j) = 1]$, of selecting $j \in [d]$ as an active coordinate, needed to compute the standard loss estimators used in adversarial multi-armed bandits (MABs) [41]. The following observation suggests that we can kernelize such loss estimators.

---

[‡]Related results were concurrently shown in [55] to compute the Hessian of a self-concordant function, which is needed to implement Newton's method.

**Observation 3.3.** *Using Theorem 2.2, we can compute the first moment $x_t$, which it turns out to provide the probabilities needed to compute the standard loss estimators used in adversarial MABs, since for any $j \in [d]$, we have that $x_t(j) := \mathbb{E}_{v \sim p_t}[\mathbb{1}\{v(j) = 1\}] = \Pr[v(j) = 1]$.*

Our algorithm (Algorithm 2 in Appendix B) kernelizes the *implicit exploration* (IX) loss estimator, $\widetilde{\ell}_t(j) = \frac{\ell_t(j)}{x_t(j)+\gamma}\mathbb{1}\{v_t(j) = 1\}$, proposed in [48], which ensures sufficient exploration for each coordinate $j \in [d]$ with low variance. Despite the fact that the implicit exploration loss estimator is biased, it satisfies the important property that, with high probability, the aggregated estimator losses are upper bounded by the realized losses plus a factor of $\widetilde{\mathcal{O}}(1/\gamma)$. The following theorem shows that the proposed algorithm is no-regret.

**Theorem 3.4** (No-Regret under Semi-Bandit Feedback). *By setting $\gamma = m/\sqrt{dT}$ and $\eta = 1/\sqrt{dT}$, Algorithm 2 achieves regret $R_T \leq \widetilde{\mathcal{O}}(m\sqrt{Td})$ with high probability.*

## 4 Efficient Kernelization in Colonel Blotto Games (CBGs)

We consider the setting proposed in [4] for the multiplayer Colonel Blotto game [17]. Each player $i \in [|\mathcal{P}|]$ must allocate $n$ soldiers among $k$ battlefields. Let variable $s_{i,h}$ denote the number of soldiers allocated by the $i$-th player to the $h$-th battlefield. Given the soldier assignments of all players, a per-battlefield loss is defined for each player $i \in [|P|]$, and and the incurred cost of player $i$ is given by the sum of her per-battlefield losses.

$\Theta(nk)$**- representation.** We aim to find succinct vector representations of each player's actions and the loss. One challenge here is that we need the action and loss representations to satisfy the definition of polyhedral games— that is, the cost of each action, given the actions of other players, must equal the dot product of the action and loss representations. One such representation is through the *layered graph* [13] (also used in [60; 61; 43]), which implies a representation dimensionality of $\Theta(n^2k)$ that can be a bottleneck for efficiently learning CCE, as shown in Table 1.

Without loss of generality, we focus on player $i$ and drop the subscript $i$. We use the notation $[b]_0 = \{0, 1, ..., b\}$ for $b \in \mathbb{N}$. Let $d = (n + 1)k$. For any action $a \in \mathcal{A}$, we consider its *succinct representation* $v \in \mathcal{V} \subset \{0, 1\}^d$ such that for all $h \in [k]$ and $s \in [n]_0$, $v[h, s] = 1$ iff $a$ assigns $s$ soldiers to the $h$-th battlefield. We similarly define the representation of the loss $\ell$, such that for all $h \in [k]$ and $s \in [n]_0$, $\ell[h, s]$ is the $h$-th battlefield loss observed when assigning $s$ soldiers to the $h$-th battlefield, given the assignments of the other players in $h$.

**Remark 4.1.** *Although the $\Theta(nk)$-representation is more straightforward to design and more succinct than the $\Theta(n^2k)$-graph-representation [61], it is not known how to derive a polytope description in the form of a small number of linear inequalities with the pure actions as corners—thus common techniques, such as Carathéodory decomposition (e.g., [26]) and barrier methods (e.g. [42; 65]) cannot be used. Kernelization overcomes this obstacle by operating directly on the game's geometry.*

**Kernelization.** With the succinct representation established above, we are now ready to describe how fast kernel computations are achieved in Colonel Blotto games.

Given weight vectors $x, y \in \{0, 1\}^d$ we define the polynomial $P_{x,y}(z)$ as follows:

$$P_{x,y}(z) := \prod_{h=1}^{k} \sum_{s=0}^{n} x[h, s]y[h, s]z^s. \tag{3}$$

The key point is that the $n$-th coefficient of $z$ in $P_{x,y}(z)$ generates kernel $K_{\mathcal{V}}(x, y)$. The main idea is as follows. To compute the $n$-th coefficient of 3, we execute a running product over the factors of the polynomial. This process involves $k$ updates of the partial product. After each update, the partial product is truncated down to degree $n$. Thus, inductively we ensure that all $k$ multiplications involve polynomials of degree at most $n$. Based on the above, we derive the following proposition.

**Proposition 4.2.** *For given $x, y \in \{0, 1\}^d$, kernel $K_{\mathcal{V}}(x, y)$ can be computed in time $\mathcal{O}(nk \log n)$.*

To construct the loss estimators of the proposed algorithms, we need to compute $d$ kernels $K_{\mathcal{V}}(C_t, \bar{e}_j)$, for $j \in [d]$, for the semi-bandit setting and $d^2$ kernels $K_{\mathcal{V}}(C_t, \bar{e}_{j,j'})$, for $j, j' \in [d]$, for the bandit

setting, as well as the kernel $K_\mathcal{V}(C_t, \mathbf{1})$. A naive approach is to use Proposition 4.2 separately for each kernel, resulting in a total kernel computation time $\mathcal{O}(n^3 k^3 \log n)$ for the bandit setting and $\mathcal{O}(n^2 k^2 \log n)$ for the semi-bandit.

We provide two algorithms, namely Algorithm 3 and 4 (Appendix G), which speed up the process of computing all required kernels by leveraging the above ideas and appropriate precomputing.

**Lemma 4.3.** *At each round $t \in [T]$, all kernels $K_\mathcal{V}(C_t, \bar{e}_j)$, for $j \in [d]$, can be computed in time $\mathcal{O}(nk \log n)$. Moreover, all kernels $K_\mathcal{V}(C_t, \bar{e}_{j,j'})$, for $j, j' \in [d]$, can be computed in time $O(n^2 k^2)$.*

Combining the above with the exact sampling procedure provided in [12] (see Algorithm 5, Appendix G.3), based on which we can calculate the required kernels of our SAMPLING procedure in time $\mathcal{O}(nk \log n)$, the per-iteration complexities for the bandit and semi-bandit are $\mathcal{O}(n^\omega k^\omega \log n)$ and $\mathcal{O}(nk \log n)$, respectively. Putting everything together, we derive the following main result.

**Theorem 4.4** (Runtime to learn $\varepsilon$-CCE). *In a Colonel Blotto game, under bandit feedback, if all players adopt Algorithm 1, then the total runtime for finding an $\varepsilon$-CCE, with high probability, is $\widetilde{\mathcal{O}}(n^{2+\omega} k^{6+\omega}/\epsilon^3)$. Under semi-bandit feedback, if all players adopt Algorithm 2 (Appendix B), then the total runtime for finding an $\varepsilon$-CCE, with high probability, is $\widetilde{\mathcal{O}}(n^2 k^4/\epsilon^2)$.*

**Remark 4.5.** *If the Colonel Blotto game is two-player zero-sum (a more traditional setting which has received much attention [14; 15; 4]), then our algorithm learns an $\epsilon$-Nash equilibrium.*

## 5 Efficient Kernelization in Graphic Matroid Congestion Games (GMCGs)

In a graphic matroid congestion game (GMCG), players compete for the edges of a connected undirected graph $G = (V, E)$, with the actions of each player being spanning trees in $G$ [64; 3; 33]. We use the incidence vector representation of actions $v \in \{0, 1\}^{|E|}$ and denote by $\mathcal{V}$ the set of all these incidence vectors. Given an action profile $(v_i, v_{-i})$, the total loss of player $i$ is the sum of the losses of the selected edges of $v_i$. Typically the cost of each edge is equal to the number of players using it but our framework can also handle arbitrary edge cost functions. Next, we will show how to efficiently compute the required kernels and perform efficient sampling in GMCGs.

**Kernelization.** Given an edge weight vector $C \in \mathbb{R}^{|E|}$ to compute the kernel $K_\mathcal{V}(C, \mathbf{1})$, we make use of the *weighted Matrix-Tree Theorem* [59; 45], which states that the value of $\sum_{T \in \mathcal{V}} \prod_{e \in T} C(e)$ equals the value of a cofactor of the weighted Laplacian $A$ of the graph, where $A_{u,u} = \sum_{e' \in E \text{ incident to } u} C(e')$ and $A_{u,v} = -C(e) \cdot \mathbb{1}\{e \in E\}$ for $u \neq v$ and edge $e = (u, v)$.

A naive approach is to use the Matrix-Tree Theorem for each kernel separately, taking total time $\mathcal{O}(|E|^2 |V|^\omega)$ for the kernel computations in the bandit and $\mathcal{O}(|E||V|^\omega)$ in the semi-bandit setting.

We provide an algorithm (see Appendix H) that reduces the amortized time per kernel computation. Notably, the Matrix-Tree Theorem holds for any cofactor of the Laplacian matrix. We leverage this property by making a strategic choice of which row and column to delete. For each edge $j \in [|E|]$ consider the Laplacian used for the computation of the kernel $K_\mathcal{V}(C, \bar{e}_j)$. The Laplacian for this kernel is constructed in the same way as the one we described above for $K_\mathcal{V}(C, \mathbf{1})$ but with the difference that $C(j)$ is set to zero. The main idea is that for each node $v \in V$, we can precompute the LU decomposition of $A_{-v,-v}$, that is the submatrix of $A$ derived by deleting row $v$ and column $v$, and then for each $j = (u, u') \in E$, we can fast compute kernel $K(C, \bar{e}_j)$ by computing the determinant of that kernel's Laplacian via recursive LU updating [56] in $\mathcal{O}(|V|^2)$. The key point in this analysis is that we can always select a submatrix of the kernel's Laplacian that only differs in one element than $A_{-u,-u}$. Similar arguments can also be used for fast computing $K_\mathcal{V}(C, \bar{e}_{j,j'})$.

**Sampling.** We provide an efficient implementation of the SAMPLING procedure of Algorithm 1 for GMCGs (see Appendix H, Algorithm 8). Our approach is based on an iterative kernelization process where we sample each coordinate incrementally. The challenge here is how to perform kernelization on the conditional action set $\mathcal{V}(j)$, for $j \in [|E|]$, induced by the so far sampled coordinates up to $j$. Interestingly, $\mathcal{V}(j)$ operates on an underlying multi-graph. The main idea is to transform this multi-graph into a meta-graph, where a meta-node merges the nodes of the $j$-th edge of the initial graph and a meta-edge accumulates the weights of parallel edges connecting the same nodes. First,

we show that it suffices to perform kernelization on the meta-graph (Proposition H.1) and, based on that, we show that our approach is efficient via an induction argument. Importantly, we derive the following lemma.

**Lemma 5.1.** *At each round $t \in [T]$, all kernels $K_{\mathcal{V}}(C_t, \bar{e}_j)$, for $j \in [|E|]$, can be computed in time $\mathcal{O}(|V|^{\omega+1} + |E||V|^2)$ and all kernels $K_{\mathcal{V}}(C_t, \bar{e}_{j,j'})$, for $j, j' \in [d]$, can be computed in time $\mathcal{O}(|E||V|^{\omega+1} + |E|^2|V|^2)$. Moreover,* SAMPLING$(\mathcal{V}, C_t)$ *can be implemented in time $\mathcal{O}(|E||V|^\omega)$.*

Putting everything together, we derive the following main result.

**Theorem 5.2** (Runtime to learn $\varepsilon$-CCE). *In a graphic matroid congestion game, under bandit feedback, if all players adopt Algorithm 1, then the total runtime for finding an $\varepsilon$-CCE, with high probability, is $\widetilde{\mathcal{O}}(|E|^3|V|^6(|V|^{\omega-1} + |E|)/\varepsilon^3)$. Under semi-bandit feedback, if all players adopt Algorithm 2 (Appendix B), then the total runtime for finding an $\varepsilon$-CCE, with high probability, is $\widetilde{\mathcal{O}}(|E|^2|V|^{2+\omega}/\varepsilon^2)$.*

## 6    Conclusion

In this paper, we focused on the problem of efficiently learning coarse correlated equilibrium (CCE) in polyhedral games via kernelization—beyond full-information feedback. In particular, we proposed kernelized no-regret learning algorithms that improve the runtime of state-of-the-art methods in three important classes of polyhedral games, namely Colonel Blotto, graphic matroid and network congestion games.

There are several important open questions for follow-up research:

- Most important of all is whether we can design an FPTAS algorithm for efficiently learning correlated equilibria (CE) in polyhedral games; a stronger equilibrium notion than CCE.

- Another interesting open question is whether we can further leverage kernelization to achieve $1/\varepsilon^2$ dependence in the bandit setting, with a better dependence on $d$ than Zimmert and Lattimore [65].

- Computing Nash equilibria in the general setting of the Colonel Blotto games we have studied in this paper is PPAD-hard, as any m-player normal-form game can be polynomially reduced to an m-player Colonel Blotto game. However, what can be said about the computational complexity of computing Nash equilibria in Colonel Blotto games with monotone piecewise-constant utility functions (e.g., see [12])?

We leave these as open questions for future work on the topic.

## Acknowledgments

Gabriele Farina was supported in part by the National Science Foundation award CCF-2443068, ONR grant N000142512296, and an AI2050 Early Career Fellowship. Ioannis Panageas was supported by NSF grant CCF-2454115. Most of this work was done while all authors were visiting Archimedes Research Unit, Athens, Greece. This work was also supported by the French National Research Agency (ANR) in the framework of the PEPR IA FOUNDRY project (ANR-23-PEIA-0003), the grant IRGA-SPICE (G7H-IRG24E90), and project MIS 5154714 of the National Recovery and Resilience Plan Greece 2.0 funded by the European Union under the NextGenerationEU Program.

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

# Contents

# A  Extended Related Work

**Online learning in games.**    The connection between no-regret learning and the computation of approximate CCE in games has been well-known since the work of Freund and Schapire [34] (see also [23]); assuming that all players use no-regret learning algorithms with regret $O\left(\sqrt{T}\right)$, the time-averaged history of joint-play consists a $O\left(\frac{1}{\sqrt{T}}\right)$-CCE. In [29], it was shown for two player zero-sum games that rate of convergence $\tilde{O}(1/T)$ can be achieved, improving the standard $O\left(1/\sqrt{T}\right)$ that can be derived from black-box regret analysis. Further improvements using the idea of optimism were shown in [50] and for general-sum games appeared in later works [57; 25; 30; 6] (see also references therein as there is a vast literature in learning in games and it is impossible to cite properly all works).

**Colonel Blotto games.**    The classical Colonel Blotto game introduced by Borel [18] dates back to 1953. Some notable works about computing NE in two-player zero-sum games include [4; 14; 15], and for learning CCE in multi-player games include [12; 43]. Moreover, [60; 61] show no-expected-regret learning in Colonel Blotto games which does not suffice for convergence to CCE. Our work improves upon previous works the runtime time to learn a CCE (see also Table 1).

**Congestion games.**    Congestion games are potential games [52] and always admit a pure Nash Equilibrium (NE); i.e, a state in which no agent has an incentive to unilaterally deviate. In full-information feedback, a long line of research studies the convergence properties to NE of game dynamics (e.g. best/better response play or no-regret). The seminal work of Takimoto and Warmuth [58], which studies online shortest paths, provides an efficient learning algorithm for network congestion games. Regarding the semi-bandit and bandit feedback settings, György et al. [35], based on [58], provide efficient algorithms for online shortest paths, which can also be applied to network congestion games. Moreover, efficient algorithms based on online gradient descent have also been established [49; 27]. The regret rate and per-iteration complexity, i.e., the total running time to reach a CCE of the aforementioned works is inferior to ours (see also Table 3). Regarding learning on spanning trees, Koo et al. [39] used the Matrix-Tree Theorem in the context of *directed* spanning trees for calculating the normalization factor of exponentiated gradient algorithm. However, their approach does not provide exact sampling, nor kernelization of bandit estimators.

**Algorithm 2:** Kernelized Algorithm based on IX under *semi-bandit feedback*

---

**Data:** $d, m, \eta > 0$ and $\gamma \in [0,1]$

1   Initialize $c_0(j) = 0$ and $C_0(j) = 0$ for all $j \in [d]$, $p_0 = [1/N, \dots, 1/N] \in \Delta(\mathcal{V}_i)$

2   **for** $t = 1, \dots, T$ **do**

3      Compute the kernels:    $K_{\mathcal{V}}(C_{t-1}, \mathbf{1})$ and $\{K_{\mathcal{V}}(C_{t-1}, \bar{e}_j)\}$ for $j \in [d]$

4      Sample an action $v_t \sim p_t$ (MWU) using $v_t = \text{SAMPLING}(\mathcal{V}, C_{t-1})$

5      Observe semi-bandit losses $\ell_t \in \mathbb{R}^d$

6      Compute the unconditional probabilities:    $x_t = \left(1 - \frac{K_{\mathcal{V}}(C_{t-1}, \bar{e}_1)}{K_{\mathcal{V}}(C_{t-1}, \mathbf{1})}, \dots, 1 - \frac{K_{\mathcal{V}}(C_{t-1}, \bar{e}_d)}{K_{\mathcal{V}}(C_{t-1}, \mathbf{1})}\right)$

7      Compute the IX loss estimators:    $\widetilde{\ell}_t(j) = \frac{\ell_t(j)}{x_t(j) + \gamma} \mathbb{1}\{v_t(j) = 1\}, \quad \forall j \in [d]$

8      Update the aggregated loss estimators:    $c_t(j) = c_{t-1}(j) + \widetilde{\ell}_t(j), \quad \forall j \in [d]$

9      Update the exponential cumulative loss estimators:    $C_t(j) = \exp\left(-\eta c_t(j)\right), \quad \forall j \in [d]$

10

11   **Procedure:** SAMPLING

12   **Input:** $\mathcal{V}, C$

13   Sample $v[1] \sim Be\left(1 - \frac{K_{\mathcal{V}}(C, \bar{e}_1)}{K_{\mathcal{V}}(C, \mathbf{1})}\right)$

14   **for** $j = 2, \dots, d$ **do**

15      Compute the kernel: $K_{\mathcal{V}(j)}$

16      Set $\mathcal{V}(j) = \{v' \in \mathcal{V} : v'[i] = v[i], \forall i \in [j-1]\}$ and $p_j = 1 - \frac{K_{\mathcal{V}(j)}(C, \bar{e}_j)}{K_{\mathcal{V}(j)}(C, \mathbf{1})}$

17      Sample $v[j] \sim Be(p_j)$

18   **Return:** $v$

---

## B   Semi-bandit No-Regret Learning: Analysis of Algorithm 2

**Lemma B.1** (Corollary 1, [48]). *Let $\gamma_t = \gamma \geq 0$ for all $t$. With probability at least $1 - \delta'$,*

$$\sum_{t=1}^{T} \left(\widetilde{\ell}_t(i) - \ell_t(i)\right) \leq \frac{\log(d/\delta')}{2\gamma}$$

*simultaneously holds for all $i \in [d]$.*

**Theorem B.2** (Theorem 3.4 restated). *For any $\delta \in (0,1)$, the sequence $v_1, \dots, v_T$ of actions played by Algorithm 2 with $\gamma = \frac{m}{\sqrt{dT}}$ and $\eta = \frac{1}{\sqrt{dT}}$ satisfies*

$$R(T) \leq \mathcal{O}\left(\left(m\sqrt{Td} + d\right) \log(d/\delta)\right),$$

*with probability at least $1 - \delta$.*

*Proof.* Let $\hat{L}_t(v) = \sum_{i \in \mathcal{V}} \hat{\ell}_t(i)$ be the loss estimator of selecting pure action $v$ at time step $t$ and $L_t(v) = \sum_{i \in \mathcal{V}} \ell_t(i)$ the corresponding true loss. Moreover, let $\hat{C}_t(v) = \sum_{t'=1}^{t} \hat{L}_{t'}(v)$ be the cumulative loss estimator of selecting pure action $v$ for the first $t$ time steps. Also, let $w_t(v) = \exp(-\eta \hat{C}_t(v))$, where $\eta$ is the learning rate of MWU, and let $W_t = \sum_{v \in \mathcal{V}} w_t(v)$, with $W_0 = |\mathcal{V}| = N$.

As in the standard analysis of MWU, we will upper and lower bound the quantity $\log \frac{W_T}{W_0}$. First, we fix a pure action $v^* \in \mathcal{V}$ and using the fact that $W_t(v) \geq w_t(v^*)$, for all $t$, we have the following lower bound:

$$\log \frac{W_T}{W_0} \geq -\eta \hat{C}_T(v^*) - \log N \tag{4}$$

On the other hand, assuming $\eta \hat{L}_t(v) \leq 1$ for all $A \in \mathcal{A}$ (this condition will be verified later), using the elementary inequalities $e^x \leq 1 + x + x^2$ for $|x| \leq 1$ and $\ln(1 + y) \leq y$ for $y > -1$, we get the standard following upper bound:

$$\log \frac{W_t}{W_{t-1}} = \log \sum_{v \in \mathcal{V}} p_t(v) \exp(-\eta \hat{L}_t(v)) \tag{5}$$

$$\leq \log \sum_{v \in \mathcal{V}} p_t(A)(1 - \eta \hat{L}_t(v) + \eta^2 \hat{L}_t(v)^2) \tag{6}$$

$$\leq \eta^2 \sum_{v \in \mathcal{V}} p_t(v) \hat{L}_t(v)^2 - \eta \sum_{v \in \mathcal{A}} p_t(v) \hat{L}_t(v) \tag{7}$$

Now, we will upper bound the first term and lower bound the second term of 7, as follows:

$$\sum_{v \in \mathcal{V}} p_t(v) \hat{L}_t(v)^2 = \sum_{v \in \mathcal{V}} p_t(v) \left( \sum_{i \in v} \hat{\ell}_t(i) \right)^2 \tag{8}$$

$$\leq m \sum_{v \in \mathcal{V}} p_t(v) \sum_{i \in v} \hat{\ell}_t(i)^2 \tag{9}$$

$$= m \sum_{i \in [d]} \hat{\ell}_t(i)^2 \sum_{v \in \mathcal{V}: i \in v} p_t(v) \tag{10}$$

$$= m \sum_{i \in [d]} x_t(i) \hat{\ell}_t(i)^2 \tag{11}$$

$$= m \sum_{i \in [d]} x_t(i) \frac{\ell_t(i) \mathbb{1}\{i \in v_t\}}{x_t(i) + \gamma} \hat{\ell}_t(i) \tag{12}$$

$$\leq m \sum_{i \in v_t} q_t(i) \hat{\ell}_t(i) \tag{13}$$

$$\leq m \sum_{i \in v_t} \hat{\ell}_t(i) \tag{14}$$

where in 9 we have used the property that the arithmetic mean is less or equal than the quadratic mean, in 13 we have used the fact that $\ell_t(i) \leq 1$ and we defined $q_t(i) = \frac{x_t(i)}{x_t(i) + \gamma}$, and in 14 we have the fact that $q_t(i) \leq 1$.

Similarly, we lower bound the second term as follows:

$$\sum_{v \in \mathcal{V}} p_t(v) \hat{L}_t(v) = \sum_{v \in \mathcal{V}} p_t(v) \sum_{i \in v} \hat{\ell}_t(i) \tag{15}$$

$$= \sum_{i \in [d]} \hat{\ell}_t(i) \sum_{v \in \mathcal{V}: i \in v} p_t(v) \tag{16}$$

$$= \sum_{i \in v_t} \frac{x_t(i)}{x_t(i) + \gamma} \ell_t(i) \tag{17}$$

$$= \sum_{i \in v_t} \ell_t(i) - \gamma \sum_{i \in v_t} \frac{\ell_t(i)}{x_t(i) + \gamma} \tag{18}$$

$$= \sum_{i \in v_t} \ell_t(i) - \gamma \sum_{i \in v_t} \hat{\ell}_t(i) \tag{19}$$

Now, summing for $t = 1, 2, ..., T$, we get:

$$\log \frac{W_T}{W_0} \le \eta^2 m \sum_{t=1}^{T} \sum_{i \in v_t} \hat{\ell}_t(i) - \eta \sum_{t=1}^{T} \sum_{i \in v_t} (\ell_t(i) - \gamma \hat{\ell}_t(i)) \tag{20}$$

Combining the above with the lower bound of 4, we get the following:

$$-\eta \hat{C}_t(v^*) - \log N \le \eta^2 m \sum_{t=1}^{T} \sum_{i \in v_t} \hat{\ell}_t(i) - \eta \sum_{t=1}^{T} \sum_{i \in v_t} (\ell_t(i) - \gamma \hat{\ell}_t(i)) \tag{21}$$

which implies that

$$\sum_{t=1}^{T} \sum_{i \in v_t} \ell_t(i) - \hat{C}_T(v^*) \le \frac{\log N}{\eta} + \eta m \sum_{t=1}^{T} \sum_{i \in v_t} \hat{\ell}_t(i) + \sum_{t=1}^{T} \sum_{i \in v_t} \gamma \hat{\ell}_t(i) \tag{22}$$

$$\Rightarrow \sum_{t=1}^{T} L_t(v_t) - \sum_{t=1}^{T} \hat{L}_t(v^*) \le \frac{\log N}{\eta} + \sum_{t=1}^{T} \sum_{i \in v_t} (\eta m + \gamma) \hat{\ell}_t(i) \tag{23}$$

Then, using Lemma B.1, with probability at least $1 - \delta'$, we get the following:

$$\sum_{t=1}^{T} L_t(v_t) - \sum_{t=1}^{T} L_t(v^*) \le \frac{m \log(d/\delta')}{2\gamma} + \frac{\log N}{\eta} + (\eta m + \gamma) \sum_{t=1}^{T} \sum_{i \in v_t} \hat{\ell}_t(i) \tag{24}$$

$$\le \frac{m \log(d/\delta')}{2\gamma} + \frac{\log N}{\eta} + (\eta m + \gamma) \sum_{i \in [d]} \sum_{t=1}^{T} \hat{\ell}_t(i) \tag{25}$$

Now, we can apply Lemma B.1 to the term $\sum_{i \in [d]} \sum_{t=1}^{T} \hat{\ell}_t(i)$. Then with probability at least $1 - 2\delta'$ we obtain:

$$\sum_{t=1}^{T} (L_t(v_t) - L_t(v^*)) \le \frac{m \log(d/\delta')}{2\gamma} + \frac{\log N}{\eta} + (\eta m + \gamma) \sum_{t=1}^{T} \sum_{i \in [d]} \ell_t(i) + (\eta m + \gamma) \frac{d \log(d/\delta')}{2\gamma} \tag{26}$$

$$\le \frac{m \log(d/\delta')}{2\gamma} + \frac{2m \log d}{\eta} + (\eta d m + \gamma d)T + (\eta m + \gamma) \frac{d \log(d/\delta')}{2\gamma} \tag{27}$$

where in the last inequality we have used the fact that $N \le \sum_{i=0}^{m} \binom{d}{i} \le m d^m \Rightarrow \log N \le 2m \log d$ and that $\sum_{i \in v_t} \ell_t(i) \le m$, which holds because $|v_t|_1 \le m$.

Next we will optimize over parameters $\gamma$ and $\eta$ to minimize the RHS in 27. We have one constraint over the parameters. In the proof of 7 we used the condition $\eta \hat{L}_t(v_t) \le 1$ for all $t \in [T]$. It is easy to verify that if $\eta m \le \gamma$ then the above condition is satisfied. We set $\gamma = \frac{m}{\sqrt{dT}}$ and $\eta = \frac{1}{\sqrt{dT}}$ to balance the dominating terms in the regret bound. Moreover, we set $\delta = 2\delta'$. Therefore, using the above and plug them in 27, we get the following:

$$\sum_{t=1}^{T} (L_t(v_t) - L_t(v^*)) \le m\sqrt{dT} \log(2d/\delta) + 2m\sqrt{dT} \log d + 2m\sqrt{dT} + d \log(2d/\delta) \tag{28}$$

Finally, we obtain our result by setting $v^* = \arg\min_{v \in \mathcal{V}} \sum_{t=1}^{T} L_t(v)$.

$\square$

## C  Second Moment Calculation via Kernelization

**Theorem C.1** (Theorem 3.1 restated). *Let $\Sigma_t(q_t) := \sum_{v \in \mathcal{V}} q_t(v)vv^T$ be the autocorrelation matrix under the law of $q_t$. Then, for all $j, j' \in [d]$, it holds that:*

$$\Sigma_t(q_t)[j,j'] = 1 - \frac{K(b^{(t)}, \bar{e}_j) + K(b^{(t)}, \bar{e}_{j'}) - K(b^{(t)}, \bar{e}_{j,j'})}{K(b^{(t)}, \mathbf{1})}$$

*Proof.* We observe that for all $j, j' \in [d]$, the feature map $\phi(\bar{e}_{j,j'})$ satisfies for all $v \in \mathcal{V}$

$$
\begin{aligned}
\phi(\bar{e}_{j,j'})[M] &= \prod_{k:v(k)=1} \bar{e}_{j,j'}[k] = \prod_{k:v(k)=1} \mathbb{1}_{k \neq j \wedge k \neq j'} = \mathbb{1}_{j \notin v \wedge j' \notin v} \\
&= 1 - \mathbb{1}_{j \in v \vee j' \in v} \\
&= 1 - \mathbb{1}_{j \in v} - \mathbb{1}_{j' \in v} + \mathbb{1}_{j,j' \in v} \\
&= \mathbb{1}_{j \notin v} + \mathbb{1}_{j' \notin v} + \mathbb{1}_{j,j' \in v} - 1
\end{aligned}
$$

Using the fact that $\phi(\mathbf{1}) = \mathbf{1}$ and $\phi(\bar{e}_j)[v] = \mathbb{1}_{j \notin v}$, we conclude that for all $j, j' \in [d], v \in \mathcal{V}$

$$\mathbb{1}_{j,j' \in v} = \phi(\mathbf{1})[v] + \phi(\bar{e}_{j,j'})[v] - \phi(\bar{e}_j)[v] - \phi(\bar{e}_{j'})[v] \tag{29}$$

Therefore, for all $j, j' \in [d]$, we obtain for the autocorrelation matrix

$$\Sigma_t(q_t)[j,j'] = \sum_{v \in \mathcal{V}} q_t(v)(vv^T)[j,j'] = \sum_{v \in \mathcal{V}} q_t(v)\mathbb{1}_{j,j' \in v} \tag{30}$$

$$= \sum_{v \in \mathcal{V}} q_t(v)(\phi(\mathbf{1})[v] + \phi(\bar{e}_{j,j'})[v] - \phi(\bar{e}_j)[v] - \phi(\bar{e}_{j'})[v]) \tag{31}$$

$$= \frac{\langle \phi(b^{(t)}), \phi(\mathbf{1}) \rangle + \langle \phi(b^{(t)}), \phi(\bar{e}_{j,j'}) \rangle - \langle \phi(b^{(t)}), \phi(\bar{e}_j) \rangle - \langle \phi(b^{(t)}), \phi(\bar{e}_{j'}) \rangle}{K(b^{(t)}, \mathbf{1})} \tag{32}$$

$$= \frac{K(b^{(t)}, \mathbf{1}) + K(b^{(t)}, \bar{e}_{j,j'}) - K(b^{(t)}, \bar{e}_j) - K(b^{(t)}, \bar{e}_{j'})}{K(b^{(t)}, \mathbf{1})} \tag{33}$$

$$= 1 - \frac{K(b^{(t)}, \bar{e}_j) + K(b^{(t)}, \bar{e}_{j'}) - K(b^{(t)}, \bar{e}_{j,j'})}{K(b^{(t)}, \mathbf{1})}, \tag{34}$$

where the third equation follows from (29), the fourth from Theorem 2.2, and the fifth from the definition of $K(\cdot, \cdot)$.

$\square$

# D Layered Graph Representation in Colonel Blotto

**Definition D.1** (Layered Graph [13]). The layered graph has $k+1$ layers and $n+1$ vertices in each layer. Let $v_{i,j}$ denote the $j$-th vertex in the $i$-th layer ($0 \leq i \leq k$ and $0 \leq j \leq n$). For any $0 \leq i \leq k$ there exists a directed edge from $v_{i-1,j}$ to $v_{i,l}$ iff $0 \leq j \leq l \leq n$.

**Lemma D.2** (Pure actions in a Layered Graph [13]). *Each directed path in the layered graph starting from $v_{0,0}$ and ending at $v_{n,k}$ is equivalent to exactly one pure action of the Colonel Blotto game, and vice versa. For each pure action, the reward for each battlefield is associated with a unique edge of the directed path.*

However, the layered graph, which has been used to succinctly represent the action space for learning in Colonel Blotto games, see [60; 61; 43], implies a representation complexity of $\Theta(n^2 k)$ that can be a bottleneck for efficient no-regret learning and convergence to CCE.

# E Barycentric Spanners

Before proceeding with the proposed algorithm for the bandit setting, we introduce the important notion of barycentric spanners [10]. We will use barycentric spanners to ensure adequate exploration of each coordinate $j \in [d]$, sufficient to guarantee low variance of the loss estimators.

**Definition E.1.** A subset of independent vectors $\{b_1, \ldots, b_d\} \subseteq \mathcal{V}$ is said to be $C$-approximate barycentric spanner of $\mathcal{V}_i$, with $C > 1$, if, for all $v \in \mathcal{V}$, there exists $\alpha \in \mathbb{R}^d$ such that

$$v = \sum_{j=1}^{d} \alpha_j b_j \quad \text{and} \quad |\alpha_j| \leq C, \quad \text{for all } j \in [d].$$

We define $B$ to be the matrix whose columns are the barycentric spanners $\{b_1, \ldots, b_d\}$.

The following proposition ensures that, if specific conditions hold, there exists an efficient algorithm for computing a $C$-approximate barycentric spanner.

**Proposition E.2** (Proposition 2.5, [9]). *Suppose $S \subseteq \mathbb{R}^d$ is a compact set not contained in any proper linear subspace. Given an oracle for optimizing linear functions over S, for any $C > 1$ there exists an algorithm that computes a $C$-approximate barycentric spanner for S in polynomial time, using $\mathcal{O}\left(d^2 \log_C(d)\right)$ calls to the optimization oracle.*

## E.1 Computing an Approximate-Barycentric Spanner for Colonel Blotto Games

**Proposition E.3** (Oracle for finding best-response in polynomial-time). *Given a reward vector $r$, the following linear optimization problem*

$$\max_{V \in \mathcal{V}_i} r^T V$$

*can be solved in time $\mathcal{O}(n^2 k)$.*

*Proof.* We will solve the following linear optimization problem (which corresponds to playing best-response with respect to the reward vector $r$):

$$\max_{V \in \mathcal{V}_i} r^T V \tag{35}$$

The above problem is equivalent to the problem of finding the longest path from a directed weighted DAG (with $|V|$ nodes and $|E|$ edges), which can be solved via Dynamic Programming in time $|V| \cdot |E|$. To do so, we leverage the Layered Graph representation (see Section D), which is a DAG with $\Theta(nk)$ nodes and $\Theta(n^2 k)$ edges. More specifically, in the Layered Graph, in layer $h \in [k]$ the edge $e_h = (u_{h,i}, u_{h+1,j})$ for $i \leq j$ and $i, j \in [n]_0$ corresponds to assigning $j - i$ soldiers on battlefield $h + 1$. On each edge, we use as edge weight the battlefield reward taken by assigning the corresponding number of soldiers on the corresponding battlefield, and the longest path of this graph,

denoted by $x^* \in \mathbb{R}^{n^2 k}$, represents the best response with respect to $r$. Thus, we can solve the linear optimization problem in time $\mathcal{O}(n^2 k)$.

The only thing left to do is to get $V^* = \arg\max_{V \in \mathcal{V}_i} r^T V$ from $x^*$. It is straightforward that there exists an one-to-one correspondence between these two vectors. To get $V^*$, one must do the following:

- Initialize $V^* = 0 \in \mathbb{R}^d$.

- For each layer $h \in [k]$, given the selected edge $e_h = (u_{h,i}, u_{h+1,j})$ in $x^*$, assign $V^*[h, j - i] = 1$.

$\square$

**Lemma E.4** (Polynomial-time algorithm for $C$-barycentric spanner in Colonel Blotto). *In Colonel Blotto, for $C > 1$, there exists a polynomial-time algorithm that computes a $C$-approximate barycentric spanner for $\mathcal{V}_i$ in time $\mathcal{O}(n^4 k^3 \log_C(nk))$.*

*Proof.* To prove Lemma E.4, first observe that $\mathcal{V}_i$ satisfies Proposition E.2 because it is compact and is not contained in any proper linear subspace of $\mathbb{R}^d$. Now, we need to have access to an oracle for optimizing linear functions over $\mathcal{V}_i$. To do so, we utilize the oracle for finding a best response given a reward vector from Proposition E.3.

Therefore, each oracle call needs time $\mathcal{O}(n^2 k)$. Now, using Proposition E.2, to compute the barycentric spanner for $\mathcal{V}_i$, one may use the algorithm defined in [9] that computes an approximate $C$-spanner (with $C > 1$) in time $\mathcal{O}(n^4 k^3 \log_C(nk))$.

### E.2 Computing an Approximate-Barycentric Spanner for Graphic Matroid Congestion Games

The idea is similar to above but using the Kruskal algorithm as the oracle. To compute a $C$-barycentric spanner here we need $\mathcal{O}(|E|^2 \log_C(|E|)$.

## F  Bandit No-Regret Learning: Analysis of Algorithm 1

We have the following:

$$\sup_{x,y\in\mathcal{V}} x^T \Sigma_t^+ y \leq \sup_{\alpha,\beta\in[-C,C]^d} \alpha^T B^T \Sigma_t^+ B\beta \tag{36}$$

$$\leq \sup_{\|\alpha\|=\|\beta\|=C\sqrt{d}} \alpha^T B^T \Sigma_t^+ B\beta \tag{37}$$

$$\leq C^2 d \cdot \|B^T \Sigma_t^+ B\|_2 \tag{38}$$

$$= C^2 d \cdot \lambda_{\max}\left(B^T \Sigma_t^+ B\right) \tag{39}$$

$$= C^2 d \cdot \frac{1}{\lambda_{\min}\left(B^{-1}\Sigma_t B^{-T}\right)} \tag{40}$$

$$= C^2 \frac{d}{\lambda_{\min}\left(B^{-1}\left(\frac{\gamma}{d}BB^T + (1-\gamma)\mathbb{E}_{\hat{p}_t}[vv^T]\right)B^{-T}\right)} \tag{41}$$

$$\leq \frac{C^2 d^2}{\gamma\lambda_{\min}\left(B^{-1}BB^T B^{-T}\right)} \tag{42}$$

$$= \frac{C^2 d^2}{\gamma} \tag{43}$$

where $B \in R^{d\times d}$ is a full rank matrix that has the approximate spanners as columns. In 40, we have used the fact that $B$ is invertible because it is full rank, and also that $\Sigma_t$ is non-singular (see [28]) which implies that $\Sigma_t^{-1}$ exists and thus the pseudo-inverse matrix $\Sigma_t^+$ equals the inverse matrix, i.e., $\Sigma_t^+ = \Sigma_t^{-1}$. Moreover, in 42, we used the Weyl's inequality.

Let $C_t$ be the autocorrelation matrix under the law of the exploration distribution on the barycentric spanner. We aim to bound the minimum non-zero eigenvalue of $C_t$. Similarly to above, we have

$$\sup_{x\in\mathcal{V}} x^T C_t^{-1} x \leq \sup_{\alpha\in[-C,C]^d} \alpha^T B^T C_t^{-1} B\alpha \tag{44}$$

$$\leq \sup_{\|\alpha\|=\|\beta\|=C\sqrt{d}} \alpha^T B^T C_t^{-1} B\alpha \tag{45}$$

$$\leq C^2 d \cdot \|B^T C_t^{-1} B\|_2 \tag{46}$$

$$= C^2 d \cdot \lambda_{\max}\left(B^T C_t^{-1} B\right) \tag{47}$$

$$= C^2 d \cdot \frac{1}{\lambda_{\min}\left(B^{-1} C_t B^{-T}\right)} \tag{48}$$

$$= C^2 \frac{d}{\lambda_{\min}\left(B^{-1}\left(\frac{1}{d}BB^T\right)B^{-T}\right)} \tag{49}$$

$$\leq C^2 d^2. \tag{50}$$

Let $\mathbb{E}_t[\cdot]$ denote expectation conditioned on the past events; i.e. the realized rewards received and the actions taken by player $i$ up to time step $t-1$. Also, let $\mathbf{1}$ be the ones vector. We define $L_t(v) = \ell_t \cdot v$, and similarly $\widehat{L}_t(v) = \widehat{\ell}_t \cdot v$. In the following analysis, we drop the superscript $i$ and sometimes write $q_t$ and $p_t$ for the distributions of player $i$. For now, we assume that $T \geq 8d^2 m$, an assumption that will be verified later by the average regret guarantee for the convergence to CCE.

Using the above analysis, along with the basic lemma of [11], we can easily get the following lemma with the basic properties of the algorithm.

**Lemma F.1.** *For any $v \in \mathcal{V}_i$ and $t \in [T]$, the following hold:*

1. *(unbiasedness)* $\mathbb{E}_t[\widehat{\ell}_t] = \ell_t$

2. $v^T \boldsymbol{\Sigma}_t^{-1} v \le \frac{d^2 C^2}{\gamma}$

3. $\left| \widehat{L}_t(v) \right| \le \frac{d^2 m C^2}{\gamma}$

4. $\mathbb{E}_t[v_t^T \boldsymbol{\Sigma}_t^{-1} v_t] = d$

5. $\mathbb{E}_t \left[ \left( \widehat{\ell}_t \cdot v \right)^2 \right] \le m^2 v^T \boldsymbol{\Sigma}_t^{-1} v$

Now, using Lemma F.1 and selecting $\eta = \frac{\gamma}{d^2 m C^2} = \frac{1}{d^{4/3} m^{2/3} C^2 T^{1/3}}$, we have

$$\left| \eta \widehat{L}_t(v) \right| \le 1.$$

**Lemma F.2** (Bernstein's inequality for martingales). *Let $Y_1$, ..., $Y_T$ be a martingale difference sequence. Suppose that $Y_t \in [a, b]$ and*

$$\mathbb{E} \left[ Y_t^2 \mid X_{t-1}, \dots, X_1 \right] \le \sigma$$

*for all $t \in \{1, \dots, T\}$. Then for all $\varepsilon > 0$,*

$$\Pr \left( \sum_{t=1}^{T} Y_t > \sqrt{2\sigma T \ln(1/\delta)} + 2\ln(1/\delta)(b-a)/3 \right) \le \delta$$

**Lemma F.3.** *Simultaneously for any $v \in \mathcal{V}_i$, with probability at least $1 - \delta$, it holds that*

$$\sum_{t=1}^{T} \left( \widehat{L}_t(v) - L_t(v) \right) \le \left( \frac{dm^{3/2} C}{\sqrt{\gamma}} + m^{3/2} \right) \sqrt{2T \ln(d/\delta)} + \frac{4}{3} \ln(d/\delta) \left( \frac{d^2 m^2 C^2}{\gamma} + m^2 \right)$$

*Proof.* Fix any $v \in \mathcal{V}_i$, we define $Y_t(v) = \widehat{L}_t(v) - L_t(v)$. $Y_t$ is a martingale difference sequence. Using Lemma F.1, the following hold:

$$\bullet \quad \sqrt{\text{Var}_t \, Y_t(v)} = \sqrt{\text{Var}_t \left[ \widehat{L}_t(v) - L_t(v) \right]} \tag{51}$$

$$\le \sqrt{\mathbb{E}_t \left[ \left( \widehat{L}_t(v) - L_t(v) \right)^2 \right]} \tag{52}$$

$$\le \sqrt{\mathbb{E}_t \left[ \left( \widehat{L}_t(v) \right)^2 \right]} + \sqrt{\mathbb{E}_t \left[ (L_t(v))^2 \right]} \tag{53}$$

$$\le m \sqrt{v^\top \boldsymbol{\Sigma}_t^{-1} v} + m \tag{54}$$

$$\le m \sqrt{\frac{d^2 C^2}{\gamma}} + m \tag{55}$$

$$= \frac{mdC}{\sqrt{\gamma}} + m \tag{56}$$

where in 53 we used the Cauchy-Schwarz inequality and in 55 we used Lemma F.1.

$$\bullet \quad |\widehat{L}_t(v) - L_t(v)| \le \frac{md^2 C^2}{\gamma} + m \tag{57}$$

Now by applying the Bernstein's inequality (Lemma F.2), with probability at least $1 - \frac{\delta}{|\mathcal{V}_i|}$ we obtain

$$\sum_{t=1}^{T} Y_t(v) \leq \left( \frac{mdC}{\sqrt{\gamma}} + m \right) \sqrt{2T \ln(|\mathcal{V}_i|/\delta)} + \frac{4}{3} \ln(|\mathcal{V}_i|/\delta) \left( \frac{md^2C^2}{\gamma} + m \right) \tag{58}$$

$$= \left( \frac{dm^{3/2}C}{\sqrt{\gamma}} + m^{3/2} \right) \sqrt{2T \ln(d/\delta)} + \frac{4}{3} \ln(d/\delta) \left( \frac{m^2d^2C^2}{\gamma} + m^2 \right) \tag{59}$$

Taking the union bound, we obtain the desired result.

$\square$

**Lemma F.4.** *With probability at least $1 - \delta$,*

$$\sum_{t=1}^{T} \sum_{v \in \mathcal{B}} \frac{\gamma}{nm} \widehat{L}_t(v) \leq \gamma mT + \left( \sqrt{\gamma} dm^{3/2}C + \gamma m^{3/2} \right) \sqrt{2T \ln(d/\delta)} + \frac{4}{3} \ln(d/\delta) \left( m^2d^2C^2 + \gamma m^2 \right)$$

*Proof.* Using Lemma F.3, with probability at least $1 - \delta$, we have, simultaneously for all $v \in \mathcal{B}$,

$$\frac{1}{d} \sum_t \gamma \widehat{L}_t(v) \leq \frac{1}{d} \left( \gamma \sum_t L_t(v) + \gamma \left( \frac{dm^{3/2}C}{\sqrt{\gamma}} + m^{3/2} \right) \sqrt{2 \ln(d/\delta)} \right.$$
$$\left. + \frac{4\gamma}{3} \ln(d/\delta) \left( \frac{d^2m^2C^2}{\gamma} + m^2 \right) \right) \tag{60}$$
$$\leq \frac{1}{d} \left( \gamma mT + \left( \sqrt{\gamma} dm^{3/2}C + \gamma m^{3/2} \right) \sqrt{2 \ln(d/\delta)} \right.$$
$$\left. + \frac{4}{3} \ln(d/\delta) \left( d^2m^2C^2 + \gamma m^2 \right) \right) \tag{61}$$

Summing over the $d$ elements of the spanner, and using the fact that $L_t(v) \leq m$, we get the result of the statement. $\square$

**Lemma F.5.** *With probability at least $1 - \delta$,*

$$\sum_{t=1}^{T} \ell_t \cdot v_t - \sum_{t=1}^{T} \sum_{v \in \mathcal{V}_i} p_t(v) \widehat{\ell}_t \cdot v \leq (m\sqrt{d} + m) \sqrt{2T \ln(1/\delta)} + \frac{4}{3} \ln(1/\delta) \left( \frac{d^2mC^2}{\gamma} + m \right).$$

*Proof.* The proof follows directly from the proof of Lemma 6 in [11], using $|Y_t| \leq \frac{d^2mC^2}{\gamma} + m$, and $\mathrm{Var}_t Y_t \leq m\sqrt{d} + m$. $\square$

**Lemma F.6.** *With probability at least $1 - \delta$,*

$$\sum_{t=1}^{T} \eta \sum_{v \in \mathcal{V}_i} p_t(v) \left( \widehat{\ell}_t \cdot v \right)^2 \leq \eta dm^2 T + \eta \left( \frac{d^2m^2C^2}{\gamma} + dm^2 \right) \sqrt{2T \ln(1/\delta)}$$

*Proof.* The proof directly follows the proof of Lemma 8 from [11], by using that the summands $v_t^T \Sigma_t^{-1} v_t$ are bounded by $\frac{d^2C^2}{\gamma}$.

$\square$

**Theorem F.7** (Theorem 3.2 restated). *For $T \geq 8d^2m$ and for any $\delta \in (0, 1)$, the sequence $v_1, \ldots, v_T$ of actions played by Algorithm 2 with $\gamma = \frac{d^{2/3}m^{1/3}}{T^{1/3}}$ and $\eta = \frac{1}{4d^{4/3}m^{2/3}T^{1/3}}$ satisfies*

$$R_T \leq \widetilde{\mathcal{O}} \left( d^{2/3}m^{4/3}T^{2/3} \right).$$

*Proof.* Following the standard analysis of MWU (also similar to our analysis in the semi-bandit setting), we have that,

$$\frac{W_{t+1}}{W_t} = \sum_{v \in \mathcal{V}_i} \frac{w_t(v) \exp\left(-\eta \widehat{L}_t(v)\right)}{W_t} \tag{62}$$

$$\leq \sum_{v \in \mathcal{V}_i} \frac{w_t(v)}{W_t} \left(1 - \eta \widehat{L}_t(v) + \eta^2 \left(\widehat{L}_t(v)\right)^2\right) \tag{63}$$

$$\leq 1 + \frac{\eta}{1-\gamma} \left(-\sum_{v \in \mathcal{V}_i} p_t(v) \widehat{L}_t(v) + \sum_{v \in \mathcal{B}} \frac{\gamma}{d} \widehat{L}_t(v) + \sum_{v \in \mathcal{V}_i} p_t(v) \eta \left(\widehat{L}_t(v)\right)^2\right) \tag{64}$$

since by definition of $p_t$,

$$\frac{w_t(v)}{W_t} = \frac{p_t(v) - \frac{\gamma}{d} \mathbb{1}\{v \in \mathcal{B}\}}{1-\gamma}.$$

Fix any $v^* \in \mathcal{V}_i$. We have that,

$$\ln\left(\frac{W_{T+1}}{W_1}\right) \geq -\eta\left(\sum_{t=1}^{T} \widehat{L}_t(v^*)\right) - \ln|\mathcal{V}_i| \tag{65}$$

$$\geq -\eta\left[\sum_{t=1}^{T} L_t(v^*) + \left(\frac{dm^{3/2}C}{\sqrt{\gamma}} + m^{3/2}\right)\sqrt{2T\ln(d/\delta)} + \frac{4}{3}\ln(d/\delta)\left(\frac{d^2m^2C^2}{\gamma} + m^2\right) + \frac{m\ln d}{\eta}\right] \tag{66}$$

$$\geq -2\eta\left[\sum_{t=1}^{T} L_t(v^*) + \left(d^{2/3}m^{4/3}CT^{2/3} + m^{3/2}\sqrt{T}\right)\sqrt{2\ln(d/\delta)}\right.$$
$$\left. + \frac{4}{3}\ln(d/\delta)\left(d^{2/3}m^{4/3}C^2T^{1/3} + m^2\right) + \frac{m\ln d}{\eta}\right] \tag{67}$$

where in 66 we used Lemma F.3, and in 67 we used the fact that $\gamma = \frac{d^{2/3}m^{1/3}}{T^{1/3}}$.

Putting these together, using Lemmas F.4, F.5 and F.6 in 64, we have

$$\frac{W_{t+1}}{W_t} \leq 1 + \frac{\eta}{1-\gamma}\left(-\sum_{t=1}^{T} L_t(u_t) + (m\sqrt{d} + m)\sqrt{2T\ln(1/\delta)} + \frac{4}{3}\ln(1/\delta)\left(\frac{d^2mC^2}{\gamma} + m\right)\right.$$
$$+ \gamma mT + \left(\sqrt{\gamma}dm^{3/2}C + \gamma m^{3/2}\right)\sqrt{2T\ln(d/\delta)} + \frac{4}{3}\ln(d/\delta)\left(d^2m^2C^2 + \gamma m^2\right)$$
$$\left. + \eta dm^2T + \eta\left(\frac{d^2m^2C^2}{\gamma} + dm^2\right)\sqrt{2T\ln(1/\delta)}\right) \tag{68}$$

Taking logs, using the fact that $\ln(1 + x) \leq x$, and also the fact that $\frac{\eta}{1-\gamma} \leq 2\eta$, because we have assumed that $T \geq 8d^2m$, and summing over $t$, we have

$$\ln\left(\frac{W_{T+1}}{W_1}\right) \le 2\eta\left(-\sum_{t=1}^{T} L_t(u_t) + (m\sqrt{d}+m)\sqrt{2T\ln(1/\delta)} + \frac{4}{3}\ln(1/\delta)\left(\frac{d^2 m C^2}{\gamma}+m\right)\right.$$

$$+ \gamma m T + \left(\sqrt{\gamma}dm^{3/2}C + \gamma m^{3/2}\right)\sqrt{2T\ln(d/\delta)} + \frac{4}{3}\ln(d/\delta)\left(d^2 m^2 C^2 + \gamma m^2\right)$$

$$\left.+ \eta d m^2 T + \eta\left(\frac{d^2 m^2 C^2}{\gamma}+dm^2\right)\sqrt{2T\ln(1/\delta)}\right) \tag{69}$$

$$= 2\eta\left(-\sum_{t=1}^{T} L_t(u_t) + (m\sqrt{d}+m)\sqrt{2T\ln(1/\delta)} + \frac{4}{3}\ln(1/\delta)\left(d^{4/3}m^{2/3}C^2 T^{1/3}+m\right)\right.$$

$$+ d^{2/3}m^{4/3}T^{2/3} + \left(d^{4/3}m^{5/3}CT^{1/3} + d^{2/3}m^{11/6}T^{1/6}\right)\sqrt{2\ln(d/\delta)}$$

$$+ \frac{4}{3}\ln(d/\delta)\left(d^2 m^2 C^2 + d^{2/3}m^{7/3}T^{1/3}\right)$$

$$\left.+ \frac{m^{2/3}T^{2/3}}{d^{1/3}} + \left(m\sqrt{T} + \frac{m^{2/3}T^{1/6}}{d^{1/3}}\right)\sqrt{2\ln(1/\delta)}\right) \tag{70}$$

where in 70 we used the definitions of $\gamma$ and $\eta$.

Finally, using 67 and 70, rearranging terms, dividing with $\eta$, using the fact that $\ln d/\eta = d^{4/3}m^{2/3}T^{1/3}\ln d$, and rescaling $\delta = 4\delta$, with probability at least $1-\delta$, simultaneously for all $u^* \in \mathcal{V}_i$, we have that,

$$\sum_{t=1}^{T}\left(L_t(v_t) - L_t(v^*)\right) \le \widetilde{\mathcal{O}}\left(d^{2/3}m^{4/3}T^{2/3}\right).$$

$\square$

## G   Kernelization in Colonel Blotto games

---

**Algorithm 3:** Efficient First-Moment Kernel Computations in Colonel Blotto games

---

**Data:** $C_t$

1 /* Compute the partial products $P_1$ and $P_2$ */

2 $P_1^{(t)}[0] = 1$

3 **for** $h = 0, ..., k - 1$ **do**

4  $\quad P_{\text{left}}(z) = P_1^{(t)}[h](z) \cdot \sum\limits_{s=0}^{n} C_t[h+1, s] \cdot z^s$

5  $\quad P_1^{(t)}[h+1](z) = $ truncate $P_{\text{left}}(z)$ to degree $n$

6 $P_2^{(t)}[0] = 1$

7 **for** $h = 0, ..., k - 1$ **do**

8  $\quad P_{\text{right}}(z) = P_2^{(t)}[h](z) \cdot \sum\limits_{s=0}^{n} C_t[h+1, s] \cdot z^s$

9  $\quad P_2^{(t)}[h+1](z) = $ truncate $P_{\text{right}}(z)$ to degree $n$

10 /* Compute the $d + 1$ kernels */

11 $K_{\mathcal{V}}(C_t, \mathbf{1}) = n$-th degree coefficient of $P_1^{(t)}[k]$

12 **for** $h = 1, ..., k$ **do**

13  $\quad P_{-h} = P_1^{(t)}[h-1](z) \cdot P_2^{(t)}[k-h](z)$

14  $\quad \sum\limits_{s=0}^{n} \alpha_s \cdot z^s = $ truncate $P_{-h}$ to degree $n$

15  $\quad$ **for** $s = 0, ..., n$ **do**

16  $\quad\quad K_{\mathcal{V}}(C_t, \bar{e}_{h,s}) = K_{\mathcal{V}}(C_t, \mathbf{1}) - \alpha_{n-s} \cdot C_t[h, s]$

---

**Algorithm 4** Efficient Second-Moment Kernel Computations in Colonel Blotto games

**Require:** $C^{(t)}$
1: # Compute the interval products
2: **for** $h, h'$ in $[k+1]_0 \times [k+1]_0$ **do**
3: $\quad P_{int}[h, h'] = 1$
4: **for** $h = 1, ..., k-1$ **do**
5: $\quad P_{int}[h, h](z) = \sum\limits_{s=0}^{n} C_{h,s}^{(t)} \cdot z^s$
6: $\quad$ **for** $h' = h, ..., k-1$ **do**
7: $\qquad P(z) = P_{int}[h, h'](z) \cdot \sum\limits_{s=0}^{n} C_{h'+1,s}^{(t)} \cdot z^s$
8: $\qquad P_{int}[h, h'+1](z) = $ truncate $P(z)$ to degree $n$
9: # $P_{int}[h, h'](z) = \prod\limits_{i=h}^{h'} \sum\limits_{s=0}^{n} C_{i,s}^{(t)} \cdot z^s$
10: # Compute the $d^2$ kernels
11: **for** $h = 1, ..., k$ **do**
12: $\quad$ # case 1: $h' = h$
13: $\quad P_{-h}(z) = P_{int}[1, h-1](z) \cdot P_{int}[h+1, k](z)$
14: $\quad \sum\limits_{s=0}^{n} \alpha_s \cdot z^s = $ truncate $P_{-h}$ to degree $n$
15: $\quad$ **for** $s = 0, ..., n$ **do**
16: $\qquad$ **for** $s' = 0, ..., n$ **do**
17: $\qquad\quad K_{\mathcal{V}}(C^{(t)}, \bar{e}_{h,h,s,s'}) = K_{\mathcal{V}}(C^{(t)}, \mathbf{1}) - \alpha_{n-s} \cdot C_{h,s}^{(t)} - \alpha_{n-s'} \cdot C_{h,s'}^{(t)} \cdot \mathbb{1}\{s \neq s'\}$
18: $\quad$ # case 2: $h' > h$
19: $\quad$ **if** $h < k$ **then**
20: $\qquad$ **for** $h' = h+1, ..., k$ **do**
21: $\qquad\quad P_h(z) = \sum\limits_{s=0}^{n} C_{h,s}^{(t)} \cdot z^s$
22: $\qquad\quad$ **for** $s = 0, ..., n$ **do**
23: $\qquad\qquad P_{h,s}(z) = P_h(z) - C_{h,s}^{(t)} \cdot z^s$
24: $\qquad\qquad P_{-h,h'}(z) = P_{int}[1, h-1](z) \cdot P_{int}[h+1, h'-1](z) \cdot P_{int}[h'+1, k](z)$
25: $\qquad\qquad P_{-h'}(z) = P_{-h,h'}(z) \cdot P_{h,s}(z)$
26: $\qquad\qquad \sum\limits_{s=0}^{n} \alpha_s \cdot z^s = $ truncate $P_{-h'}$ to degree $n$
27: $\qquad\qquad$ **for** $s' = 0, ..., n$ **do**
28: $\qquad\qquad\quad K_{\mathcal{V}}(C^{(t)}, \bar{e}_{h,h',s,s'}) = K_{\mathcal{V}}(C^{(t)}, \mathbf{1}) - \alpha_{n-s'} \cdot C_{h',s'}^{(t)}$

## G.1 Proof of Proposition 4.2

**Proposition G.1.** *For given $x, y \in \{0, 1\}^d$, there exists an algorithm that computes the kernel $K(x, y)$ in time $\mathcal{O}(nk \log n)$.*

*Proof.* To compute the $n$-th coefficient of 3, we execute a running product over the factors of the polynomial. This process involves $k$ updates of the partial product. After each update, the partial product is truncated down to degree $n$. Thus, inductively we ensure that all $k$ multiplications involve polynomials of degree at most $n$. Each multiplication can be implemented with FFT [20] in $\mathcal{O}(n \log n)$ time. The overall complexity over the $k$ multiplications is $\mathcal{O}(nk \log n)$ and after the truncated product is computed the target coefficient is obtained in $\mathcal{O}(1)$. $\qquad\square$

## G.2 Proof of Lemma 4.3

The proof is based on Algorithm 3. We define the running product $P_l^{(t)}[i]$ from left to right, which is the sum of the degree $0$ to $n$ terms of the polynomial $\prod_{i'=1}^{i} \sum_{j=0}^{n} C_t[i', j] \cdot z^j$. We can compute all polynomials $P_l^{(t)}[i]$, for $i = 1, ..., k$ in total time $nk \log n$ using the following induction argument:

Given $P_l^{(t)}[i]$, we compute $P_l^{(t)}[i+1]$ by performing the polynomial multiplication $P_l^{(t)}[i](z) \cdot \sum_{j=0}^{n} C_t[i+1, j] \cdot z^j$ and truncating all terms of degree greater than $n$. The two multiplied polynomials have degree $n$, so the multiplication can be done in time $n \log n$ using FFT, while the truncation of the higher degree terms can be done in time $n$ since the product polynomial has degree $2n$. Repeating this procedure for $i = 1, ..., k-1$ we get all left-to-right partial products in total time $nk \log n$.

Similarly, we define the running product $P_r^{(t)}[i]$ from right to left as the sum of the degree $0$ to $n$ terms of the polynomial $\prod_{i'=k-i+1}^{k} \sum_{j=0}^{n} C_t[i', j] \cdot z^j$. Similarly to $P_l^{(t)}$, we can compute all right-to-left partial products $P_r^{(t)}[i]$, for $i = 1, ..., k$ in total time $nk \log n$.

Now, using the above partial products, we compute all the kernels required for (O)MWU at time step $t$. All polynomial multiplications in the Algorithm are performed using FFT so that each of them takes time $n \log(n)$.

Following similar logic as above, via Algorithm 4 we get the desired result.

$\square$

## G.3 Alternative Proof of Lemma 4.3

Efficient sampling of MWU in CBGs has been studied in Beaglehole et al. [12]. A useful tool for this purpose is the partition function defined in equation 100. Here we describe their method with details and we extend their ideas to the efficient calculation of first and second order moments of the MWU distribution.

**Remark G.2.** *We give an algorithm (Algorithm 4) that performs the second moment computation in terms of kernels. The algorithm follows a similar logic to Algorithm 3, but is somewhat more complicated, due to the nature of the problem. In steps where polynomial multiplication is performed, we imply that the multiplication is implemented efficiently through FFT.*

We remind that the calculation of first order moments was used in our method for learning in the semi-bandit setting and the second moments appear in the calculation of the autocorrelation matrix which is used in the bandit setting. Next we proceed to the technical details of our methods.

Focusing on a single player, at time step $t$, let $\ell_t[h, s]$ be the loss observed by the player when assigning $s$ soldiers to the $h$-th battlefield, given the assignments of the other players in this battlefield. Moreover, let

$$c_h^{(t)}(s) = \sum_{\tau=1}^{t} \ell_t[h, s]$$

We define the partition function

$$f_h(y) = \sum_{x_1 + \cdots + x_h = y} \prod_{i=1}^{h} \exp\left(-\eta c_i^{(t)}(x_i)\right). \tag{71}$$

We also define the partition function $g_h(y)$, which is similar to $f_h(y)$ but aggregates battlefields in the reverse order.

$$g_h(y) = \sum_{x_h + \cdots + x_k = y} \prod_{i=h}^{k} \exp\left(-\eta c_i^{(t)}(x_i)\right). \tag{72}$$

Let $L^{(t)}(x_1, ..., x_k)$ be the cumulative loss at timestep $t$. $L^{(t)}$ can be decomposed into the cumulative losses per battlefield as follows:

$$L^{(t)}(x_1, ..., x_k) = \sum_{h=1}^{k} c_h^{(t)}(x_h) \tag{73}$$

Under MWU the probability of some assignment $x_1, ..., x_k$ at timestep $t$ can be written as

$$\Pr[s_1, ..., s_k = x_1, ..., x_k] \propto \exp\left(-\eta L^{(t)}(x_1, ..., x_k)\right) \tag{74}$$

$$= \exp\left(-\eta \sum_{h=1}^{k} c_h^{(t)}(x_h)\right) \tag{75}$$

Marginal probabilities of soldier assignments at a single battlefield can be written as follows:

$$\Pr[s_k = s] = \sum_{x_1+\cdots+x_{k-1}=n-s} \Pr[s_1, ..., s_{k-1}, s_k = x_1, ..., x_{k-1}, s] \tag{76}$$

$$\propto \sum_{x_1+\cdots+x_{k-1}=n-s} \exp\left(-\eta \sum_{h=1}^{k-1} c_h^{(t)}(x_h) - \eta c_k^{(t)}(s)\right) \tag{77}$$

$$= \exp\left(-\eta c_k^{(t)}(s)\right) \sum_{x_1+\cdots+x_{k-1}=n-s} \exp\left(-\eta \sum_{h=1}^{k-1} c_h^{(t)}(x_h)\right) \tag{78}$$

$$= \exp\left(-\eta c_k^{(t)}(s)\right) \sum_{x_1+\cdots+x_{k-1}=n-s} \prod_{h=1}^{k-1} \exp\left(-\eta c_h^{(t)}(x_h)\right) \tag{79}$$

$$= \exp\left(-\eta c_k^{(t)}(s)\right) \cdot f_{k-1}(n-s) \tag{80}$$

Moreover, we can compute the conditional probability of each soldier assignment at a single battlefield, given a set of soldier assignments at other battlefields:

$$\Pr\{s_{k-h} = s \mid s_{k-h+1}, \ldots, s_k\} = \sum_{\substack{x_1+\cdots+x_{k-h-1}= \\ n-s-\sum_{j=k-h+1}^{k} s_j}} \Pr[s_1, ..., s_{k-h-1}, s_{k-h} = x_1, ..., x_{k-h-1}, s \mid s_{k-h+1}, \ldots, s_k] \tag{81}$$

$$\propto \sum_{\substack{x_1+\cdots+x_{k-h-1}= \\ n-s-\sum_{j=k-h+1}^{k} s_j}} \exp\left(-\eta \sum_{j=1}^{k-h-1} c_j^{(t)}(x_j) - \eta c_{k-h}^{(t)}(s) - \eta \sum_{j=k-h+1}^{k} c_j^{(t)}(s_j)\right) \tag{82}$$

$$= \exp\left(-\eta c_{k-h}^{(t)}(s) - \eta \sum_{j=k-h+1}^{k} c_j^{(t)}(s_j)\right) \sum_{\substack{x_1+\cdots+x_{k-h-1}= \\ n-s-\sum_{j=k-h+1}^{k} s_j}} \exp\left(-\eta \sum_{j=1}^{k-h-1} c_j^{(t)}(x_j)\right) \tag{83}$$

$$= \exp\left(-\eta c_{k-h}^{(t)}(s) - \eta \sum_{j=k-h+1}^{k} c_j^{(t)}(s_j)\right) f_{k-h-1}\left(n - \left(\sum_{j=k-h+1}^{k} s_j\right) - s\right) \tag{84}$$

$$\propto \exp\left(-\eta c_{k-h}^{(t)}(s)\right) \cdot f_{k-h-1}\left(n - \left(\sum_{j=k-h+1}^{k} s_j\right) - s\right) \tag{85}$$

Similarly, in terms of the partition function $g_h(y)$, we derive

$$\Pr[s_1 = s] \propto \exp\left(-\eta c_1^{(t)}(s)\right) \cdot g_{k-1}(n-s) \tag{86}$$

$$\Pr\{s_h = s \mid s_1, \ldots, s_{h-1}\} \propto \exp\left(-\eta c_h^{(t)}(s)\right) \cdot g_{k-h-1}\left(n - \left(\sum_{j=1}^{h-1} s_j\right) - s\right) \tag{87}$$

The conditional probabilities can be used to implement an efficient sampling procedure for the MWU distribution, as was proposed in [12]. For completeness we write the algorithm below.

---

**Algorithm 5** Sampling from the MWU distibution in Colonel Blotto games

---

**Require:** Soldiers $n \geq 0$, battlefields $k \geq 1$ and cumulative loss $c_h^{(t)}(s)$ for $h, s \in [k] \times [n]_0$

1: $f_0(s) = 1$ for all $s \in [n]_0$

2: **for** $h = 1, \ldots, k-1$ **do**

3:  Using FFT, calculate the convolution $(a * b)(s)$ where $a(s) = \exp\left(-\eta c_h^{(t)}(s)\right)$ and $b(s) = f_{h-1}(s)$, $s \in [n]_0$.

4:  $\forall\, s \in [n]_0$, calculate the partition function for battlefield $h$:

$$f_h(s) = \sum_{s'=0}^{s} \exp\left(-\eta c_h^{(t)}(s')\right) \cdot f_{h-1}(s - s') = (a * b)(s),$$

5: Sample the number $s_k$ of soldiers at the last battlefield:

$$\Pr[s_k = s] \propto \exp\left(-\eta c_k^{(t)}(s)\right) \cdot f_{k-1}(n - s), \ s \in [n]_0$$

6: **for** $h = 1, \ldots, k-1$ **do**

7:  Sample the number $s_{k-h}$ of soldiers at battlefield $k - h$ given the numbers of soldiers, $s_{k-h+1}, \ldots, s_k$, assigned to battlefields $k - h + 1, \ldots, k$ as follows:

$$\Pr\{s_{k-h} = s \mid s_{k-h+1}, \ldots, s_k\} \propto \exp\left(-\eta c_{k-h}^{(t)}(s)\right) \cdot f_{k-h-1}\left(n - \left(\sum_{j=k-h+1}^{k} s_j\right) - s\right)$$

for $s \in \left[n - \sum_{j=k-h+1}^{k} s_j\right]_0$

---

**Remark G.3.** *Algorithm 5 implements the* SAMPLING *procedure of Algorithms 1 and 2. The key point is that instead of explicitly calculating the required kernels, it directly computes the conditional probabilities via a partition function, which corresponds to kernelizing the conditional polytope.*

The unconditional marginals that constitute the first moment are calculated as follows:

$$\Pr[s_h = s] = \sum_{\substack{\sum_{j \neq h} x_j = n-s}} \Pr[(s_1, ..., s_h, ..., s_k) = (x_1, ..., s, ..., x_k)] \tag{88}$$

$$\propto \sum_{\substack{\sum_{j \neq h} x_j = n-s}} \exp\left(-\eta \sum_{j \neq h} c_j^{(t)}(x_j) - \eta c_h^{(t)}(s)\right) \tag{89}$$

$$= \exp\left(-\eta c_h^{(t)}(s)\right) \sum_{s'=0}^{n-s} \sum_{\substack{\sum_{j=1}^{h-1} x_j = s'}} \exp\left(-\eta \sum_{j=1}^{h-1} c_j^{(t)}(x_j)\right) \sum_{\substack{\sum_{j=h+1}^{k} x_j = n-s-s'}} \exp\left(-\eta \sum_{j=h+1}^{k} c_j^{(t)}(x_j)\right) \tag{90}$$

$$= \exp\left(-\eta c_h^{(t)}(s)\right) \sum_{s'=0}^{n-s} f_{h-1}(s') g_{h+1}(n - s - s') \tag{91}$$

$$= \exp\left(-\eta c_h^{(t)}(s)\right) (f_{h-1} * g_{h+1})(n - s) \tag{92}$$

We can precompute the partition functions $f_h(y), g_h(y)$, for all $h \in [k]$ and $y \in [n]_0$ in total time $kn \log n$ utilizing the self reducible structure of the partition function (see algorithm 6 lines 1-5 for details). Then we compute $f_{h-1} * g_{h+1}$ for all $h \in [k]$ and $y \in [n]_0$ in total time $kn \log n$ with FFT. Using these calculations each term $\Pr[s_h = s]$ computation takes constant time. Note that this method is essentially equivalent to the kernel method we describe in the main paper (Algorithm 3).

For the calculation of the second-order marginals that constitute the second moment, we will make use of the interval partition function $f_{h,h'}(y)$, that aggregates possible assignments between the $h$ and the $h'$ battlefields.

$$f_{h,h'}(y) = \sum_{x_h + \cdots + x_{h'} = y} \prod_{i=h}^{h'} \exp\left(-\eta c_i^{(t)}(x_i)\right). \tag{93}$$

$$\Pr[s_h, s_{h'} = s, s'] = \sum_{\substack{\sum_{j \neq h, h'} x_j = n-s-s'}} \Pr[(s_1, ..., s_h, ..., s_{h'}, ..., s_k) = (x_1, ..., s, ..., s', ..., x_k)] \tag{94}$$

$$\propto \sum_{\substack{\sum_{j \neq h, h'} x_j = n-s-s'}} \exp\left(-\eta \sum_{j \neq h, h'} c_j^{(t)}(x_j) - \eta c_h^{(t)}(s) - \eta c_{h'}^{(t)}(s')\right) \tag{95}$$

$$= \exp\left(-\eta c_h^{(t)}(s)\right) \exp\left(-\eta c_{h'}^{(t)}(s')\right) \sum_{x=0}^{n-s-s'} \sum_{y=0}^{n-s-s'-x} \sum_{\substack{\sum_{j=1}^{h-1} x_j = x}} \exp\left(-\eta \sum_{j=1}^{h-1} c_j^{(t)}(x_j)\right)$$
$$\sum_{\substack{\sum_{j=h+1}^{h'-1} x_j = y}} \exp\left(-\eta \sum_{j=h+1}^{h'-1} c_j^{(t)}(x_j)\right) \sum_{\substack{\sum_{j=h'+1}^{k} x_j = n-s-s'-x-y}} \exp\left(-\eta \sum_{j=h'+1}^{k} c_j^{(t)}(x_j)\right) \tag{96}$$

$$= \exp\left(-\eta c_h^{(t)}(s)\right) \exp\left(-\eta c_{h'}^{(t)}(s')\right) \sum_{x=0}^{n-s-s'} \sum_{\substack{\sum_{j=1}^{h-1} x_j = x}} \exp\left(-\eta \sum_{j=1}^{h-1} c_j^{(t)}(x_j)\right)$$

$$\sum_{y=0}^{n-s-s'-x} \sum_{\substack{\sum_{j=h+1}^{h'-1} x_j = y}} \exp\left(-\eta \sum_{j=h+1}^{h'-1} c_j^{(t)}(x_j)\right) \sum_{\substack{\sum_{j=h'+1}^{k} x_j = n-s-s'-x-y}} \exp\left(-\eta \sum_{j=h'+1}^{k} c_j^{(t)}(x_j)\right)$$

$$\tag{97}$$

$$= \exp\left(-\eta c_h^{(t)}(s)\right) \exp\left(-\eta c_{h'}^{(t)}(s')\right) \sum_{x=0}^{n-s-s'} f_{1,h-1}(x)$$

$$\sum_{y=0}^{n-s-s'-x} f_{h+1,h'-1}(y) f_{h'+1,k}(n-s-s'-x-y) \tag{98}$$

$$= \exp\left(-\eta c_h^{(t)}(s)\right) \exp\left(-\eta c_{h'}^{(t)}(s')\right) \sum_{x=0}^{n-s-s'} f_{1,h-1}(x)(f_{h+1,h'-1} * f_{h'+1,k})(n-s-s'-x)$$

$$= \exp\left(-\eta c_h^{(t)}(s)\right) \exp\left(-\eta c_{h'}^{(t)}(s')\right) (f_{1,h-1} * (f_{h+1,h'-1} * f_{h'+1,k}))(n-s-s')$$

$$\tag{99}$$

We observe that the marginal probabilities only depend on the interval partition function and the cumulative loss per battlefield. We can precompute the partition function $f_{h,h'}(y)$, for all $h, h' \in [k]^2 : h \le h'$ and $y \in [n]_0$ in total time $nk^2 \log n$ utilizing the self reducible structure of the partition function (see Algorithm 5). Then we compute the convolutions $(f_{1,h-1} * f_{h+1,h'-1} * f_{h'+1,k})(y)$ for all $h, h' \in [k]^2 : h < h'$ and $y \in [n]_0$ in total time $nk^2 \log n$ with FFT. Using these calculations each term $\Pr[s_h, s_{h'} = s, s']$ computation takes constant time.

### G.4 Proof of Theorem 4.4

*Proof.* Using Lemma 4.3 and the exact sampling procedure provided in [12] (see Algorithm 5), based on which we can calculate the required kernels of our SAMPLING procedure in time $\mathcal{O}(nk \log n)$, the per-iteration complexity for the bandit and semi-bandit algorithms is $\mathcal{O}(n^\omega k^\omega \log n)$ and $\mathcal{O}(nk \log n)$, respectively. By combining Theorems 3.2 and 3.4 with Theorem 2.1, we can achieve the desired results.

$\square$

## G.5 Similar Techniques for Efficient Implementation of Kernelized GEOMETRICHEDGE in m-sets

---

**Algorithm 6** Sampling the MWU distribution in m-sets

---

**Require:** Soldiers $n \geq 0$, battlefields $d \geq 1$ and cumulative loss $c_h^{(t)} \cdot b$ for $h, s \in [d] \times [n]_0$

1: $f_0(y) = 1$ for all $y \in [m]_0$
2: **for** $h = 1, ..., d - 1$ **do**
3:    $\forall\, y \in [m]_0$, calculate the partition function for item $h$:

$$f_h(y) = \exp\left(-\eta c_h^{(t)}\right) \cdot f_{h-1}\left(y - 1\right) + f_{h-1}\left(y\right),$$

4: Sample the selection of the last item:

$$\Pr[v_d = b] \propto \exp\left(-\eta c_d^{(t)} \cdot b\right) \cdot f_{d-1}(m - b),\ b \in \{0, 1\}$$

5: **for** $h = 1, ..., d - 1$ **do**
6:    Sample the selection $v_{d-h}$ of item $d - h$ given the selections, $v_{d-h+1}, \ldots, v_d$, of items $d - h + 1, \ldots, d$ as follows:

$$\Pr\{v_{d-h} = b \mid v_{d-h+1}, \ldots, v_d\} \propto \exp\left(-\eta c_{d-h}^{(t)} \cdot b\right) \cdot f_{d-h-1}\left(m - \left(\sum_{j=d-h+1}^{d} v_j\right) - b\right)$$

$$\text{for } b \in \left\{0, \min\left(1, m - \sum_{j=d-h+1}^{d} v_j\right)\right\}$$

---

**Summary:** *We can apply similar techniques to efficiently compute the second moment used in Algorithm 1 for the classic m-sets setting. In particular, our approach requires time $\widetilde{\mathcal{O}}(md^2)$, improving upon the DAG formulation approach of [24; 58] which requires time $\mathcal{O}(m^2 d^2)$.*

A classic setting in combinatorial bandits which is also considered in [32] are m-sets, where actions are selections of $m$ out of $d$ items. Its binary representation the action set can be written as $\mathcal{V} = \{v \in \{0, 1\}^d \mid \sum_i v_i = m\}$.

We will show how to perform efficient exact sampling and autocorrelation matrix calculation in m-sets. For this purpose we will use the partition function defined in equation 100, similarly to Blotto. The partition function resembles the kernels used in kernelized MWU. Next we proceed to the technical details of our methods.

At time step $t$, let $\ell_t[i]$ be the loss observed by the player when selecting the $i$-th item. Moreover, let

$$c_i^{(t)} = \sum_{\tau=1}^{t} \ell_t[i]$$

be the cumulative loss of the $i$-th item over the first $t$ time steps. We define the partition function

$$f_h(y) = \sum_{x_1 + \cdots + x_h = y} \prod_{i=1}^{h} \exp\left(-\eta c_i^{(t)} \cdot x_i\right) \tag{100}$$

where $x_i \in \{0, 1\}, h \in [d]$ and $y \in [m]$.
We also define the partition function $g_h(y)$, which is similar to $f_h(y)$ but aggregates the set items in the reverse order.

$$g_h(y) = \sum_{x_h + \cdots + x_d = y} \prod_{i=h}^{d} \exp\left(-\eta c_i^{(t)} \cdot x_i\right). \tag{101}$$

Let $L^{(t)}(x_1, ..., x_d)$ be the cumulative loss at timestep $t$. $L^{(t)}$ can be decomposed into the cumulative losses per item as follows:

$$L^{(t)}(x_1, ..., x_d) = \sum_{h=1}^{d} c_h^{(t)} \cdot x_h \tag{102}$$

Under MWU the probability of some assignment $x_1, ..., x_d$ at timestep $t$ can be written as

$$\Pr[v_1, ..., v_d = x_1, ..., x_d] \propto \exp\left(-\eta L^{(t)}(x_1, ..., x_d)\right) \tag{103}$$

$$= \exp\left(-\eta \sum_{h=1}^{d} c_h^{(t)} \cdot x_h\right) \tag{104}$$

Marginal probabilities over assignments can be written as follows:

$$\Pr[v_d = b] = \sum_{x_1 + \cdots + x_{d-1} = m-b} \Pr[v_1, ..., v_{d-1}, v_d = x_1, ..., x_{d-1}, b] \tag{105}$$

$$\propto \sum_{x_1 + \cdots + x_{d-1} = n-b} \exp\left(-\eta \sum_{h=1}^{d-1} c_h^{(t)} \cdot x_h - \eta c_d^{(t)} \cdot b\right) \tag{106}$$

$$= \exp\left(-\eta c_d^{(t)} \cdot b\right) \sum_{x_1 + \cdots + x_{d-1} = m-b} \exp\left(-\eta \sum_{h=1}^{d-1} c_h^{(t)} \cdot x_h\right) \tag{107}$$

$$= \exp\left(-\eta c_d^{(t)} \cdot b\right) \sum_{x_1 + \cdots + x_{d-1} = m-b} \prod_{h=1}^{d-1} \exp\left(-\eta c_h^{(t)} \cdot x_h\right) \tag{108}$$

$$= \exp\left(-\eta c_d^{(t)} \cdot b\right) \cdot f_{d-1}(m - b) \tag{109}$$

Conditional probabilities over assignments are calculated as follows:

$$\Pr\{v_{d-h} = b \mid v_{d-h+1}, \ldots, v_d\} = \sum_{\substack{x_1 + \cdots + x_{d-h-1} = \\ m-b-\sum_{j=d-h+1}^{d} v_j}} \Pr[v_1, ..., v_{d-h-1}, v_{d-h} = x_1, ..., x_{d-h-1}, b \mid v_{d-h+1}, \ldots, v_d]$$

$$\tag{110}$$

$$\propto \sum_{\substack{x_1 + \cdots + x_{d-h-1} = \\ m-b-\sum_{j=d-h+1}^{d} v_j}} \exp\left(-\eta \sum_{j=1}^{d-h-1} c_j^{(t)} \cdot x_j - \eta c_{d-h}^{(t)} \cdot b - \eta \sum_{j=d-h+1}^{d} c_j^{(t)} \cdot v_j\right)$$

$$\tag{111}$$

$$= \exp\left(-\eta c_{d-h}^{(t)} \cdot b - \eta \sum_{j=d-h+1}^{d} c_j^{(t)} \cdot v_j\right) \sum_{\substack{x_1 + \cdots + x_{d-h-1} = \\ m-b-\sum_{j=d-h+1}^{d} v_j}} \exp\left(-\eta \sum_{j=1}^{d-h-1} c_j^{(t)} \cdot x_j\right)$$

$$\tag{112}$$

$$= \exp\left(-\eta c_{d-h}^{(t)} \cdot b - \eta \sum_{j=d-h+1}^{d} c_j^{(t)} \cdot v_j\right) f_{d-h-1}\left(m - \left(\sum_{j=d-h+1}^{d} v_j\right) - b\right)$$

$$\tag{113}$$

$$\propto \exp\left(-\eta c_{d-h}^{(t)} \cdot b\right) \cdot f_{d-h-1}\left(m - \left(\sum_{j=d-h+1}^{d} v_j\right) - b\right)$$

$$\tag{114}$$

Similarly we derive

$$\Pr[v_1 = b] \propto \exp\left(-\eta c_1^{(t)} \cdot b\right) \cdot g_{d-1}(m-b) \tag{115}$$

$$\Pr\{v_h = b \mid v_1, \ldots, v_{h-1}\} \propto \exp\left(-\eta c_h^{(t)} \cdot b\right) \cdot g_{d-h-1}\left(m - \left(\sum_{j=1}^{h-1} v_j\right) - b\right) \tag{116}$$

The conditional probabilities can be used to implement an efficient sampling procedure for the MWU distribution. We write the algorithm below.

For the calculation of second-order marginals we will make use of the interval partition function $f_{h,h'}(y)$, that aggregates possible assignments between the $h$ and the $h'$ battlefields.

$$f_{h,h'}(y) = \sum_{x_h + \cdots + x_{h'} = y} \prod_{i=h}^{h'} \exp\left(-\eta c_i^{(t)}(x_i)\right). \tag{117}$$

$$\Pr\left[v_h, v_{h'} = b, b'\right] = \sum_{\substack{\sum\limits_{j \neq h,h'} x_j = m-b-b'}} \Pr[(v_1, ..., v_h, ..., v_{h'}, ..., v_d) = (x_1, ..., b, ..., b', ..., x_d)]$$
$$\tag{118}$$

$$\propto \sum_{\substack{\sum\limits_{j \neq h,h'} x_j = m-b-b'}} \exp\left(-\eta \sum_{j \neq h,h'} c_j^{(t)} \cdot x_j - \eta c_h^{(t)} \cdot b - \eta c_{h'}^{(t)} \cdot b'\right) \tag{119}$$

$$= \exp\left(-\eta c_h^{(t)} \cdot b\right) \exp\left(-\eta c_{h'}^{(t)} \cdot b'\right) \sum_{x=0}^{m-b-b'} \sum_{y=0}^{m-b-b'-x} \sum_{\substack{\sum\limits_{j=1}^{h-1} x_j = x}} \exp\left(-\eta \sum_{j=1}^{h-1} c_j^{(t)} \cdot x_j\right)$$

$$\sum_{\substack{\sum\limits_{j=h+1}^{h'-1} x_j = y}} \exp\left(-\eta \sum_{j=h+1}^{h'-1} c_j^{(t)} \cdot x_j\right) \sum_{\substack{\sum\limits_{j=h'+1}^{d} x_j = m-b-b'-x-y}} \exp\left(-\eta \sum_{j=h'+1}^{d} c_j^{(t)} \cdot x_j\right)$$
$$\tag{120}$$

$$= \exp\left(-\eta c_h^{(t)} \cdot b\right) \exp\left(-\eta c_{h'}^{(t)} \cdot b'\right) \sum_{x=0}^{m-b-b'} \sum_{\substack{\sum\limits_{j=1}^{h-1} x_j = x}} \exp\left(-\eta \sum_{j=1}^{h-1} c_j^{(t)} \cdot x_j\right)$$

$$\sum_{y=0}^{m-b-b'-x} \sum_{\substack{\sum\limits_{j=h+1}^{h'-1} x_j = y}} \exp\left(-\eta \sum_{j=h+1}^{h'-1} c_j^{(t)} \cdot x_j\right) \sum_{\substack{\sum\limits_{j=h'+1}^{d} x_j = m-b-b'-x-y}} \exp\left(-\eta \sum_{j=h'+1}^{d} c_j^{(t)} \cdot x_j\right)$$
$$\tag{121}$$

$$= \exp\left(-\eta c_h^{(t)} \cdot b\right) \exp\left(-\eta c_{h'}^{(t)} \cdot b'\right) \sum_{x=0}^{m-b-b'} f_{1,h-1}(x)$$

$$\sum_{y=0}^{m-b-b'-x} f_{h+1,h'-1}(y) f_{h'+1,d}(m-b-b'-x-y) \tag{122}$$

$$= \exp\left(-\eta c_h^{(t)} \cdot b\right) \exp\left(-\eta c_{h'}^{(t)} \cdot b'\right) \sum_{x=0}^{m-b-b'} f_{1,h-1}(x)(f_{h+1,h'-1} * f_{h'+1,d})(m-b-b'-x)$$
$$\tag{123}$$

$$= \exp\left(-\eta c_h^{(t)} \cdot b\right) \exp\left(-\eta c_{h'}^{(t)} \cdot b'\right) (f_{1,h-1} * (f_{h+1,h'-1} * f_{h'+1,d}))(m - b - b') \tag{124}$$

We observe that the marginal probabilities only depend on the interval partition function and the cumulative loss per battlefield. We can precompute the partition function $f_{h,h'}(y)$, for all $h, h' \in [d]^2 : h \leq h'$ and $y \in [n]_0$ in total time $md^2 \log m$ utilizing the self reducible structure of the partition function (see algorithm 6 lines 1-5 for details). Then we compute the convolutions $(f_{1,h-1} * f_{h+1,h'-1} * f_{h'+1,d})(y)$ for all $h, h' \in [d]^2 : h < h'$ and $y \in [n]_0$ in total time $md^2 \log m$ with FFT. Using these calculations, each term $\Pr[v_h, v_{h'} = b, b']$ computation takes constant time.

We remind that the autocorrelation matrix, which is used to construct the loss estimator in GEOMETRICHEGDE, has the probabilities $\Pr[v_h, v_{h'} = b, b']$ as entries. At this point we have shown how to efficiently sample from MWU and how to compute the autocorrelation matrix and thus that GEOMETRICHEGDE can be efficiently implemented.

### G.5.1   Comparison with a DAG approach

One can easily see that online learning in **m-sets** can be modeled as online path planning in an appropriately constructed DAG with $E = \mathcal{O}(d * m)$ edges. In this graph, nodes are parameterized by two indices $i, j \in [d+1] \times [m]$. The source is node $N(0,0)$ and the sink is $N(d+1, m)$. At node $N(i, j)$, $1 \leq i \leq d$ we have considered items 1 to $i-1$ and we have selected $j$ of them. If we select item $i$ we make a transition from $N(i, j)$ to $N(i+1, j+1)$, otherwise we make a transition to $N(i+1, j)$. Transitions that lead to selecting more than $m$ items are illegal and at the sink node $N(d+1, m)$ we should have selected exactly $m$ items. This way, there is an equivalence between paths in the constructed DAG and selections of $m$ out of $d$ items and in both cases the reward is linear to the components.

For m-sets over $d$ items the corresponding DAG has $|E| = \Theta(md)$ edges. Then, sampling can be performed through weight pushing [58] in $\mathcal{O}(E) = \mathcal{O}(md)$, which is similar to complexity of sampling via the partition function. For the calculation of the autocorrelation matrix the approach of path planning in the m-set DAG would need $\mathcal{O}(E^2) = \mathcal{O}(m^2 d^2)$ using the techniques of [58]. Compared to the above our approach saves an $m$ factor. Thus, along with partitions (which is the action set in Blotto) m-sets is another application, where kernelization is beneficial compared to standard techniques such as weight pushing.

# H  Kernelization in Graphic Matroid Congestion Games

---

**Algorithm 7:** Efficient First-Moment Kernel Computations in Graphic Matroid Congestion games

**Data:** $C \in \mathbb{R}^E$

1  /* Compute the weighted Laplacian $A \in \mathbb{R}^{|V| \times |V|}$ */

2  $A[u,v] = \begin{cases} \sum_{e \in E \text{ incident to } u} C(e) & \text{if } u = v \\ -C(e) & \text{if } e = (u,v) \in E \\ 0 & \text{otherwise.} \end{cases}$

3  /* Compute the LU decompositions of the submatrices $A_{-u,-u}$ */

4  **for** $u = 1, ..., |V|$ **do**

5       $A_{-u,-u} = $ the submatrix of $A$ derived by deleting row $u$ and column $u$

6       Compute the LU decomposition $(L_u, U_u)$ of $A_{-u,-u}$, that is lower triangular $L_u$ and upper triangular $U_u$ such that $A_{-u,-u} = L_u \cdot U_u$

7  /* Compute the $d$ kernels */

8  **for** $j = 1, ..., |E|$ **do**

9       Let $u_j, v_j$ be the two nodes connected by edge $j$

10      Let $E_{-j} = E \setminus \{j\}$ be the subgraph that does not have edge $j$

11      /* Compute the weighted Laplacian $A^{(j)}$ in the subgraph where edge $j$ is missing */

12      $A^{(j)}[u,v] = \begin{cases} \sum_{e \in E_{-j} \text{ incident to } u} C(e) & \text{if } u = v \\ -C(e) & \text{if } e = (u,v) \in E_{-j} \\ 0 & \text{otherwise.} \end{cases}$

13      $A^{(j)}_{-u_j,-u_j} = $ the submatrix of $A^{(j)}$ derived by deleting row $u_j$ and column $u_j$

14      Compute the LU decomposition $(L, U)$ of $A^{(j)}_{-u_j,-u_j}$ in $\mathcal{O}(|V|^2)$ using the precomputed matrices $L_{u_j}, U_{u_j}$ and the technique of [56]

15      Compute the kernel $K_{\mathcal{V}}(C, \bar{e}_j) = \det(L \cdot U)$ in $\mathcal{O}(|V|^2)$

---

## H.1  Proof of Proposition H.1

Let $\bar{G} = (\bar{\mathcal{V}}, \bar{E})$ be a connected multigraph, and let $\hat{G} = (\hat{\mathcal{V}}, \hat{E})$ be the *meta-graph* associated with $\bar{G}$, defined as follows:

1. The vertex sets coincide:
$$\hat{\mathcal{V}} = \bar{\mathcal{V}}.$$

2. For an edge $e = (u, v) \in \bar{E}$ with weight $\bar{w}(e)$:

   - If $e$ is the unique edge between $u$ and $v$, then $e \in \hat{E}$ with the same weight:
   $$\hat{w}(e) = \bar{w}(e).$$

   - Otherwise, let $\{e'\} \subset \bar{E}$ be all parallel edges connecting $u$ and $v$. Then, we define a single *meta-edge* $\hat{e} \in \hat{E}$ with weight equal to the total weight of the merged edges:
   $$\hat{w}(\hat{e}) = \sum_{e' \in \{e'\}} \bar{w}(e').$$

     We say that $\hat{e}$ is a *merged meta-edge*, and write $e' \subset \hat{e}$ if the edge $e' \in \bar{E}$ participates in the construction of $\hat{e} \in \hat{E}$.

3. Let $\bar{\mathcal{T}}$ denote the set of spanning trees of $\bar{G}$, and let $\hat{\mathcal{T}}$ denote the set of spanning trees of $\hat{G}$.

We derive the following proposition.

**Proposition H.1.** *It holds that* $K_{\hat{\mathcal{V}}}(\hat{w}, \mathbf{1}) = K_{\bar{\mathcal{V}}}(\bar{w}, \mathbf{1})$.

*Proof.* It holds that:

$$K_{\hat{\mathcal{V}}}(\hat{w}, \mathbf{1}) = \sum_{\hat{\mathcal{T}} \in \hat{\mathcal{V}}} \prod_{\hat{e} \in \hat{\mathcal{T}}} \hat{w}(\hat{e}) \tag{125}$$

$$= \sum_{\hat{\mathcal{T}} \in \hat{\mathcal{V}}} \prod_{e \in \hat{\mathcal{T}}: e \text{ not merged}} \hat{w}(e) \prod_{\hat{e} \in \hat{\mathcal{T}}: \hat{e} \text{ merged}} \hat{w}(\hat{e}) \tag{126}$$

$$= \sum_{\hat{\mathcal{T}} \in \hat{\mathcal{V}}} \prod_{e \in \hat{\mathcal{T}}: e \text{ not merged}} \bar{w}(e) \prod_{\hat{e} \in \hat{\mathcal{T}}: \hat{e} \text{ merged}} \sum_{e' \subset \hat{e}} \bar{w}(e') \tag{127}$$

$$= \sum_{\bar{\mathcal{T}} \in \bar{\mathcal{V}}} \prod_{\bar{e} \in \bar{\mathcal{T}}} \bar{w}(\bar{e}) \tag{128}$$

$$= K_{\bar{\mathcal{V}}}(\bar{w}, \mathbf{1}) \tag{129}$$

$\square$

## H.2 Proof of Lemma 5.1

*Proof.* **Kernelization.**

The above algorithm (Algorithm 7) shows how to compute the first-moment kernel computations in Graphic Matroid Congestion games. The time complexity and correctness of the algorithm are discussed below.

We will use the following:

- We need $\mathcal{O}(|V|^{\omega})$ time for computing an LU decomposition.

- We need $\mathcal{O}(|V|^{\omega+1})$ time for precomputing the LU decompositions of the minors.

We leverage the property of the Matrix-Tree Theorem which allows us to use any submatrix to compute the determinant of the Laplacian matrix. Therefore, we can always make a strategic choice of which row and column to delete. For each edge $j \in [|E|]$ consider the Laplacian used for the computation of the kernel $K_{\mathcal{V}}(C, \bar{e}_j)$. The Laplacian for this kernel is constructed in the same way as the one we described above for $K_{\mathcal{V}}(C, \mathbf{1})$ but with the difference that $C(j)$ is set to zero. For each node $v \in V$, we precompute the LU decomposition of the minors $A_{-v,-v}$—that is the submatrix of $A$ derived by deleting row $v$ and column $v$—and then for each $j = (u, u') \in E$, we fast compute kernel $K(C, \bar{e}_j)$ by computing the determinant of that kernel's Laplacian via recursive LU updating [56] in $\mathcal{O}(|V|^2)$. The latter is due to the fact that we can always select a submatrix of the kernel's Laplacian that only differs in one element from $A_{-u,-u}$, so we can apply the techniques of [56]. Similar arguments can also be used for fast computing $K_{\mathcal{V}}(C, \bar{e}_{j,j'})$ to derive the desired results.

**Per-Iteration complexity of SAMPLING.**

We implement the SAMPLING procedure of Algorithms 1 and 2 based on the above algorithm (Algorithm 8). Since we have guaranteed that the SAMPLING procedure performs exact sampling from a MWU($\mathcal{V}, C$), what remains to prove is that the implementation we propose correctly computes the conditional kernels. We will prove this using an induction argument on the iterations $j \in [|E|]$ of the algorithm. We will show that the algorithm correctly computes the new Bernoulli probability $p_{j+1}$.

- **Basis**: The meta-graph is initialized as the initial graph. From Theorem 2.2 and Observation 3.3, we get the unconditional probability $p_1$. The algorithm samples $v(1) \sim p_1$. If the first edge $(u, v)$ is not selected then the algorithm removes it from the new meta-graph and computes $K_{\mathcal{V}(2)}$ via the cofactor of the Laplacian of the new meta-graph. If the first edge is selected then: (a) if the first edge is *not a merging meta-edge* (that is, nodes $u$

---

**Algorithm 8:** Efficient Exact Sampling of MWU in Graphic Matroids

---

**Data:** $C \in \mathbb{R}^{|E|}$

1   **Initialize** Meta-Graph $= G(V, E)$ and assign weight $C(e)$ to each edge $e$

2   Compute kernels $K_{\mathcal{V}}(C, \bar{e}_1)$ and $K_{\mathcal{V}}(C, \mathbf{1})$ using the Matrix-Tree Theorem

3   Sample $v(1) \sim Be\left(1 - \frac{K_{\mathcal{V}}(C, \bar{e}_1)}{K_{\mathcal{V}}(C, \mathbf{1})}\right)$

4   Initialize the cumulative weight $w = 1$

5   **for** $j = 2, ..., d$ **do**

6      **if** $v(j-1) = 0$ **then**

7         Find the meta-edge of Meta-Graph containing edge $j-1$ and reduce its weight by $C(j-1)$

8      **else**

9         Find the meta-edge $e$ of Meta-Graph containing edge $j-1$

10        Update $w = w \cdot \text{weight}(e)$

11        Merge the two meta-nodes connected by the meta-edge $e$.

12        If parallel edges are created then merge them into a single meta-edge containing all the parallel edges and assign to the new meta edge weight equal to the sum of the weights of the parallel edges

13      /* Compute the kernel $K_{\mathcal{V}(j)}$ */

14      Compute a cofactor $c$ of the Laplacian of the Meta-Graph

15      $K_{\mathcal{V}(j)}(C, \mathbf{1}) = w \cdot c$

16      Find the meta-edge of Meta-Graph containing the edge $j$ and reduce its weight by $C(j)$

17      Compute a cofactor $c'$ of the Meta-Graph Laplacian using the weights of the meta-edges

18      $K_{\mathcal{V}(j)}(C, \bar{e}_j) = w \cdot c'$

19      Find the meta-edge of Meta-Graph containing the edge $j$ and increase its weight by $C(j)$

20      $p_j = 1 - \frac{K_{\mathcal{V}(j)}(C, \bar{e}_j)}{K_{\mathcal{V}(j)}(C, \mathbf{1})}$

21      Sample $v(j) \sim Be(p_j)$

---

and $v$ do not have common neighbors in the meta-graph) then the algorithm removes this edge from the graph, merges the two associated nodes of this edge in the new meta-graph, updates the cumulative weight with the weight of this edge, and computes $K_{\mathcal{V}(2)}$, (b) if the first edge is a *merging meta-edge* (i.e., nodes $u$ and $v$ do have common neighbors in the meta-graph), then the algorithm makes the above steps, but now the meta-graph is a multi-graph. In this case, the algorithm also merges the resulted parallel edges connecting the associated nodes into a meta-edge with weight equal with the sum the weights of the merged edges. The computation of $K_{\mathcal{V}(2)}$ is correct, due to Proposition H.1, because the kernel computation on a multi-graph (that is, the meta-graph after merging the nodes $u$ and $v$, but before merging the resulted parallel edges) equals the kernel computation on the corresponding new meta-graph.

- **Induction Step**: We use similar arguments with the basis, with the only difference when removing an edge. Now, if edge $j$ is not selected by the Bernoulli distribution $p_j$ but $j$ is part of a merged meta-edge (i.e., a meta-edge consisting of many edges of the initial graph), then the algorithm removes its weight from this meta-edge and computes the new meta-graph. Again the computation of $K_{\mathcal{V}(j+1)}$ is correct due to Proposition H.1.

Therefore, $\text{SAMPLING}(\mathcal{V}, C_t)$ can be implemented in time $\mathcal{O}(|E||V|^\omega)$, where $|V|^\omega$ is due to the time we need to compute a single kernel.

$\square$

### H.3   Proof of Theorem 5.2

*Proof.* We directly derive the statement of the theorem by combining Theorems 3.2 and 3.4 with Theorem 2.1 and Lemma 5.1. $\square$

# I Kernelization in Network Congestion Games

We consider the setting used in [49; 27]; that is, the network congestion game takes place on a DAG, consisting of nodes $V$ and edges $E$, and thus an action of each player is a path of a DAG. We assume that the maximal path length is $K$. Following [24], we represent an action of each player $i \in [|\mathcal{P}|]$, as the incidence vector $v \in \{0, 1\}^{|E|}$ of the corresponding path: for all $j \in [|E|]$, $v(j) = 1$ if and only if the corresponding edge is present in the path. We denote the action set (i.e., a set of path vectors) of player $i \in [|\mathcal{P}|]$ by $\mathcal{V}_i$. Given an action profile $(v_i, v_{-i})$ the total loss of player $i$, $L_i$, is the sum of the losses of the selected edges of $v_i$. Based on the above, it is easy to check that a network congestion game is a combinatorial game with $|\mathcal{P}|$ players, actions sets $\{\mathcal{V}_i\}$ and losses $\{\mathcal{L}_i\}$, where the action vectors are $|E|$-dimensional and their $L_1$-norm is at most $K$.

To perform efficient sampling in DAGs and compute all kernels needed by Algorithms 2 and 1, we utilize the methodology based on DP developed by [58]. For sampling we need time $\mathcal{O}(|E|)$. For the first moment calculation we need time $\mathcal{O}(|E|)$, while for the second moment calculation we need time $\mathcal{O}(|E|^2)$. Using also the fact that we can compute a 2-approximate barycentric spanner in time $\widetilde{\mathcal{O}}((|E| + |V|)^3)$, we obtain the following CCE convergence results.

**Theorem I.1** (Semi-bandit Convergence to CCE). *In a network congestion game, under the semi-bandit online learning setup, if all players adopt Algorithm 2, then after $\widetilde{\mathcal{O}}(|E|^{1+\omega} K^2/\epsilon^2)$ runtime, with $T \geq |E| K^2/\varepsilon^2$, the time-average joint actions, $\sigma^* := \frac{1}{T} \sum_{t=1}^{T} v_1^{(t)} \otimes \cdots \otimes v_{|\mathcal{P}|}^{(t)}$, forms an $\varepsilon$-CCE of the game with high probability.*

**Theorem I.2** (Bandit Convergence to CCE). *In a network congestion game, under the bandit online learning setup, if all players adopt Algorithm 1, then after $\widetilde{\mathcal{O}}(|E|^{2+\omega} K^4/\epsilon^3)$ runtime, with $T \geq |E|^2 K^4/\varepsilon^3$, the time-average joint actions, $\sigma^* := \frac{1}{T} \sum_{t=1}^{T} v_1^{(t)} \otimes \cdots \otimes v_{|\mathcal{P}|}^{(t)}$, forms an $\varepsilon$-CCE of the game with high probability.*

# J Efficient Uniform Random Path Sampling from a DAG

In this section, we describe a method for efficiently and exactly sampling paths uniformly at random from a Directed Acyclic Graph (DAG). This process is essential for the initialization phase of MWU. We present the algorithm's pseudocode and analyze its computational complexity as well as its correctness.

---
**Algorithm 9** Uniform Random Path Sampling from a DAG

---
**Require:** A DAG $G = (V, E)$, source node $s$, target node $t$
**Ensure:** A uniformly random path $P$ from $s$ to $t$
1: /* Path Count Precomputation: */
2: Perform a topological sort of the nodes in $G$.
3: Set $C(v) \leftarrow 0$ for all $v \in V$, and $C(t) \leftarrow 1$.
4: **for** each node $v$ in reverse topological order **do**
5:    $C(v) \leftarrow \sum_{(v,u) \in E} C(u)$
6: /* Path Sampling: */
7: Initialize $P \leftarrow [s]$ and set $v \leftarrow s$.
8: **while** $v \neq t$ **do**
9:    Calculate probabilities $P(u) \leftarrow \frac{C(u)}{\sum_{(v,w) \in E} C(w)}$ for all $(v, u) \in E$.
10:    Select the next node $u$ based on probabilities $P(u)$.
11:    Add $u$ to $P$ and update $v \leftarrow u$.
12: **return** $P$

---

**Computationally Complexity.** The precomputation step requires $\mathcal{O}(V + E)$ for the topological sort and another $\mathcal{O}(V + E)$ for calculating the path counts. Therefore, the overall complexity of the precomputation step is $\mathcal{O}(V + E)$. During the sampling phase, there are at most $\mathcal{O}(V)$ iterations, one for each node in the path. Computing transition probabilities takes $\mathcal{O}(\deg(v))$ for each node $v$, leading to a total of $\mathcal{O}(E)$ operations. Thus, the complexity of the sampling phase is $\mathcal{O}(E)$.

Combining both steps, the total complexity of the algorithm is $\mathcal{O}(V + E)$.

**Correctness Proof.** To demonstrate correctness, we prove that every path $P$ from $s$ to $t$ is selected with equal probability.

The dynamic programming step calculates $C(v)$, the number of paths from node $v$ to $t$. Using the recurrence relation:

$$C(v) = \sum_{(v,u)\in E} C(u),$$

we ensure that $C(s)$ represents the total number of paths from $s$ to $t$, and $C(v)$ indicates the number of paths passing through $v$. At each node $v$, the transition probability to a neighboring node $u$ is:

$$P(u \mid v) = \frac{C(u)}{\sum_{(v,w)\in E} C(w)} = \frac{C(u)}{C(v)}.$$

For any path $P = s \to v_1 \to v_2 \to \cdots \to t$ in $G$ the probability of selecting it is given by the product of transition probabilities:

$$P(P) = P(v_1 \mid s) \cdot P(v_2 \mid v_1) \cdot \cdots \cdot P(t \mid v_K).$$

By substituting $P(u \mid v) = \frac{C(u)}{C(v)}$, we get:

$$P(P) = \frac{C(v_1)}{C(s)} \cdot \frac{C(v_2)}{C(v_1)} \cdot \cdots \cdot \frac{C(t)}{C(v_K)} = \frac{C(t)}{C(s)} = \frac{1}{C(s)}.$$

Since $C(s)$ equals the total number of paths from $s$ to $t$, every path is selected with an equal probability of $\frac{1}{C(s)}$.

