# OpenReview forum: "Efficient Kernelized Learning in Polyhedral Games beyond Full Information: From Colonel Blotto to Congestion Games"
_NeurIPS.cc/2025/Conference — NeurIPS 2025 poster_

### Official Review · Reviewer_x1xn · 2025-06-21

**Clarity:** 3
**Significance:** 3
**Originality:** 3
**Rating:** 5
**Confidence:** 3

**Summary:**

This paper studies no-regret learning and approximate Coarse Correlated Equilibria (CCE) computation under different information settings. The authors demonstrate that efficient approaches for adversarial linear bandits can be applied to several classes of prominent games. These settings include Congestion and Colonel Blotto games, which have garnered significant attention over many decades across disciplines. In particular, the authors introduce Kernelized Geometric Hedge to achieve significant, though polynomial, improvements in runtime and representation costs. To implement their algorithm efficiently, the authors demonstrate that the second-moment of the MWU distribution can also be kernelized, and the authors design appropriate representations and kernel functions for the games they consider. For Colonel Blotto, this involved computing a generating polynomial for this kernel.

**Questions:**

What about Nash Equilibria in two player settings? Can these results be applied the obtain stronger results there?

What prevents kernelized approaches from being applied to Correlated Equilibria computation? Are these limitations fundamental?

Can you explain the semi-bandit feedback setting for your representation in Colonel Blotto? Why does semi-bandit feedback make sense in this setting?

In Beaglehole et al. 2023, they perform a piece-wise approximation of a partition function to achieve polylogarithmic dependence on the number of soldiers, can a similar trick be applied here?

**Ethical Concerns:**

["NO or VERY MINOR ethics concerns only"]

**Final Justification:**

This paper develops efficient algorithms for important problems in online learning and algorithmic game theory. I therefore recommend acceptance.

**Limitations:**

Yes.

**Paper Formatting Concerns:**

None.

**Quality:**

4

**Strengths And Weaknesses:**

Strengths:\
-	The authors address fundamental and well-studied challenges in (algorithmic) game theory and online learning.\
-	Their approach significantly improves over previous approaches, especially in bandit and semi-bandit settings.

Weaknesses:\
-	The aspects of their method that are novel, as opposed to being applications of existing ideas is unclear. This is a small weakness given the improvements they achieved on important problems. The paper may benefit from clarifying which technical ideas are adapted from other words and which are more ‘original’. \
-	The runtimes still appear somewhat large, the paper might benefit from a minimal experiment demonstrating the practicality of their proposed algorithm.

Other Minor comments:\
-	The Kernelized MWU formula (184-193) is a bit difficult to understand. This part could benefit from some additional explanation given how crucial it is to the techniques in the paper. For example, the first moment calculation theorem seems unmotivated at that position.\
-	Is there a type in Eqn 3? Why is there a j exponent over z?\
-	It would be nice to have some explanation for how this polynomial in Eqn 3 is used to compute the kernels.

---

> ### Author Rebuttal · Authors · 2025-07-30
>
> Dear reviewer,
>
> Thank you for your time, and your positive evaluation! We reply to your remarks and questions below:
>
> > The aspects of their method that are novel, as opposed to being applications of existing ideas is unclear. This is a small weakness given the improvements they achieved on important problems. The paper may benefit from clarifying which technical ideas are adapted from other works and which are more ‘original’.
>
> While our work builds on the general idea of kernelization from prior literature, our main novelty lies in extending it to previously unexplored settings—both in terms of feedback models (semi-bandit and bandit) and game structures (such as Colonel Blotto and Matroid Congestion Games).
>
> In particular, our main technical novelties are as follows:
>
> 1. Kernelization in the (semi-)bandit setting: We show how to use kernelization not only for first-moment calculations, as in prior work on full-information feedback [1], but also to efficiently compute second moments and perform exact sampling. These operations are essential in the (semi-)bandit setting and, to the best of our knowledge, had not been addressed in earlier work.
> 2. We utilize novel kernelization techniques to classes of polyhedral games that had not been analyzed from this perspective before:
> (a) In Graphic Matroid Congestion Games, we design novel algorithms and analysis (see Appendix H) for exact sampling and moment computation, resolving a long-standing open problem posed by Cesa-Bianchi & Lugosi [4] on how to perform exact sampling from the ComBand algorithm (and thus MWU) in spanning trees. In particular, the kernels used for sampling are on the conditional polytope, that is on subsets of the original actions where some coordinates are restricted to have a specific 0-1 value. Computing such conditional kernels presents additional challenges compared to standard kernels: while standard kernels can be obtained by directly applying the Matrix Tree Theorem as a black box, conditional kernels require several non-trivial technical steps before the theorem can be applied.
> (b) In Colonel Blotto games, we introduce an efficient action representation of size $nk$, develop a kernelization procedure in this representation, and apply acceleration techniques to reduce the amortized kernel computation time to $O(\log n)$. This kernelization also resolves an open question from [5]. Our techniques share similarities with those in [1] for the kernelization of m-sets, in the sense that, in both settings, a kernel can be computed via specific coefficients of some polynomial (see Eq. 3 for Colonel Blotto); however, designing fast kernelization algorithms in Colonel Blotto is substantially more challenging because the polynomial factors are $n$-degree polynomials as opposed to the binomials used for m-sets (see also Algorithm 4, Appendix G).
>
>
> > The runtimes still appear somewhat large, the paper might benefit from a minimal experiment demonstrating the practicality of their proposed algorithm.
>
> The way that OpenReview has been set up this year, it is not possible to include any simulations during the rebuttal stage (or even the result thereof through an anonymized picture or github link). However, we will be happy to take advantage of the extra page to include proof-of-concept experiments aimed at validating the efficiency of our approach—specifically, measuring $\mathrm{Reg}(T)/T$ as a function of $T$ until convergence to an ε-CCE.
>
> In more detail, we will add the following experimental setups:
> 1. Experiments on congestion games under (semi-)bandit feedback. We will use the Sioux Falls dataset (part of the well-known TransportationNetworks collection), which has been widely used in the literature to evaluate regret-like performance in congestion games. The dataset represents a network with 24 nodes and 76 edges. We will use 100 players and the standard linear cost function per edge.
> 2. Experiments on Colonel Blotto games under (semi-)bandit feedback. We will use different seeds to produce randomized Colonel Blotto games consisting of 10 players, 100 battlefields and 1000 soldiers per player. The reward function will be defined as "Winner Takes All" (that is, the battlefield reward of player $i$ is 1 if player $i$ assigned more soldiers in this battlefield than all other players, otherwise 0).
>
> >  The Kernelized MWU formula (184-193) is a bit difficult to understand. This part could benefit from some additional explanation given how crucial it is to the techniques in the paper. For example, the first moment calculation theorem seems unmotivated at that position.
>
> Thanks a lot for this comment. The key idea, as shown in [1], is that computing the first moment is sufficient to simulate MWU under full-information feedback. This is because, in the full-information setting of [1], each agent plays a mixed strategy - which is exactly this first moment - rather than sampling an action from the MWU distribution. Moreover, this first moment encapsulates sufficient information about the MWU distribution, because, using standard Carathéodory decomposition techniques, it can be efficiently decomposed into a low-support distribution, from which sampling becomes efficient. We will use the extra space to clarify this part of the preliminaries in the camera-ready version.
>
> > Is there a typo in Eqn 3? Why is there a j exponent over z?
>
> Yes, it is indeed a typo: j should have been s. Thanks!
>
> > It would be nice to have some explanation for how this polynomial in Eqn 3 is used to compute the kernels.
>
> Sure, we will include some more explanation on this in the camera-ready version.
>
> > What about Nash Equilibria in two player settings? Can these results be applied the obtain stronger results there?
>
> Yes, our results can be applied to learning Nash equilibria in the case of two-player zero-sum polyhedral games (e.g., two-player Colonel Blotto), due to equilibrium collapse. We will make sure to include a remark in the camera-ready version.
>
> > What prevents kernelized approaches from being applied to Correlated Equilibria computation? Are these limitations fundamental?
>
> In learning CE, the situation gets much more challenging. Actually, it is a major open question in the field whether one can design FPTAS algorithms to find CE in polyhedral games. Only a PTAS algorithm is known due to the recent results of [2,3]. This is actually a learning approach and it depends on efficient MWU implementation in the game's action space. Thus, kernelization can be used in this framework and yield efficient (PTAS) algorithms for learning CE in polyhedral games.
>
> > Can you explain the semi-bandit feedback setting for your representation in Colonel Blotto? Why does semi-bandit feedback make sense in this setting?
>
> In the semi-bandit setting of the Colonel Blotto game, in each battlefield, a player observes only the reward resulting from the number of soldiers they allocated to that battlefield. This setup is more realistic than the full-information setting, as in many practical scenarios players cannot access the payoffs of allocations they did not commit. For example, in real-world resource allocation tasks, feedback is typically limited to the outcomes of the specific allocations made, and obtaining counterfactual rewards for unrealized allocations is often infeasible.
>
> > In Beaglehole et al. 2023, they perform a piece-wise approximation of a partition function to achieve polylogarithmic dependence on the number of soldiers, can a similar trick be applied here?
>
> That's a great question! The technique in [5] exploits a particular structural property of the utility function, that is increasing q-piece-wise constant. In contrast, we make no such assumptions, and our result applies to general (linear) utility functions. Actually, we did explore this direction as well, but the approach encountered several obstacles. A key challenge is that, to enable learning, we need an efficient representation of each player's action and reward vectors such that the reward remains linear in the action vector. Unfortunately, a $k \log n$ representation does not seem to be able to achieve this linearity. We therefore view this as a very interesting direction for future work.
>
> ---
>
> Thank you again for your constructive input and encouraging words—please do not hesitate to reach out if you have any further questions!
>
> Kind regards,
>
> The authors
>
> ---
>
> ### References
>
> [1] Farina, Gabriele, et al. "Kernelized multiplicative weights for 0/1-polyhedral games: Bridging the gap between learning in extensive-form and normal-form games." International Conference on Machine Learning. PMLR, 2022.
>
> [2] Dagan, Yuval, et al. "From external to swap regret 2.0: An efficient reduction for large action spaces." Proceedings of the 56th Annual ACM Symposium on Theory of Computing. 2024.
>
> [3] Peng, Binghui, and Aviad Rubinstein. "Fast swap regret minimization and applications to approximate correlated equilibria." Proceedings of the 56th Annual ACM Symposium on Theory of Computing. 2024.
>
> [4] Cesa-Bianchi, Nicolo, and Gábor Lugosi. "Combinatorial bandits." Journal of Computer and System Sciences 78.5 (2012): 1404-1422.
>
> [5] Beaglehole, Daniel, et al. "Sampling equilibria: Fast no-regret learning in structured games." Proceedings of the 2023 Annual ACM-SIAM Symposium on Discrete Algorithms (SODA). Society for Industrial and Applied Mathematics, 2023.

---

> > ### Comment · Reviewer_x1xn · 2025-08-04
> > **Response**
> >
> > Thank you for your thorough reply. I will keep my positive score.

---

### Official Review · Reviewer_CYJo · 2025-07-01

**Clarity:** 3
**Significance:** 4
**Originality:** 3
**Rating:** 5
**Confidence:** 2

**Summary:**

This paper develops efficient payoff-based learning algorithms for computing coarse correlated equilibria (CCE) in polyhedral games, which have exponentially large action spaces and combinatorial structure. These include Colonel Blotto and congestion games. This paper extends kernelization techniques from full-information setting to partial-information regimes (bandit and semi-bandit feedback), enabling no-regret learning with improved runtime and regret bounds. The authors proposed efficient loss estimation and sampling methods in two games: Colonel blotto and congestion games, addressing key open problems in the literature.

**Questions:**

See weakness

**Ethical Concerns:**

["NO or VERY MINOR ethics concerns only"]

**Final Justification:**

Thanks for the response. I will maintain my score.

**Limitations:**

See weakness

**Quality:**

3

**Strengths And Weaknesses:**

Strengths
1. The paper is clearly written, and the theoretical results and proofs appear sound and rigorous.
2. It proposes efficient kernel-based algorithms for learning coarse correlated equilibria in Colonel Blotto and congestion games, achieving several state-of-the-art runtime guarantees. Notably, the work addresses key open problems in the literature, including extending kernelization techniques to the bandit setting and improving regret bounds under partial information.

Weaknesses
It is unclear whether the obtained runtime bounds are tight. The authors do not provide matching lower bounds or discuss potential optimality, leaving open the question of whether further improvements are possible.

---

> ### Author Rebuttal · Authors · 2025-07-30
>
> Dear reviewer,
>
> Thank you for your time, and your positive evaluation! We reply to your remarks and questions below:
>
> > It is unclear whether the obtained runtime bounds are tight. The authors do not provide matching lower bounds or discuss potential optimality, leaving open the question of whether further improvements are possible.
>
> To the best of our knowledge, there are no lower bounds for the specific setting we consider, that is runtime to learn CCE. In terms of regret, for the full-info case, the regret is polylogarithmic in terms of $T$ and thus near-optimal (see [1] for more details). However, in the semi-bandit and bandit case it is a major open question whether it is possible to improve on the $\sqrt{T}$, which is optimal for adversarial learning. In fully adversarial semi-bandit problems, our regret is indeed optimal in terms of $d$ and $T$ and only a factor of $\sqrt{m}$ away from the lower bound for expected regret. Nevertheless, our paper focuses on realised regret which is a stronger notion than the ordinary expected regret, so it is not clear if this is a gap that can actually be closed in this case. In the adversarial bandit case, the situation is much more complicated because the minimax optimal regret bound of $\sqrt{dT \log(|A|)}$ is currently only known to be achieved by inefficient algorithms (e.g., see [2]), which have exponential (in $d$) per-iteration complexity.
>
>
> ---
>
> Thank you again for your constructive input and encouraging words - please do not hesitate to reach out if you have any further questions!
>
> Kind regards,
>
> The authors
>
> ---
>
> ### References
> [1] Farina, Gabriele, et al. "Kernelized multiplicative weights for 0/1-polyhedral games: Bridging the gap between learning in extensive-form and normal-form games." International Conference on Machine Learning. PMLR, 2022.
>
> [2] Zimmert, Julian, and Tor Lattimore. "Return of the bias: Almost minimax optimal high probability bounds for adversarial linear bandits." Conference on Learning Theory. PMLR, 2022.

---

> > ### Comment · Reviewer_CYJo · 2025-08-05
> >
> > Thanks for the response. I will maintain my score.

---

### Official Review · Reviewer_wspx · 2025-07-02

**Clarity:** 3
**Significance:** 3
**Originality:** 3
**Rating:** 5
**Confidence:** 4

**Summary:**

The authors consider the problem of computing coarse correlated equilibria for games that have succinct representation. In particular, they focus on polyhedral games, including network congestion games, Colonet Blotto games. Building upon a recent development that enables to “kernelize” no-regret like dynamics, the authors are able to provide algorithms that provably converge to a CCE and that do so requiring an amount of computation that depends polynomially on the size of the succinct game (as opposed to polynomially in the size of the expanded game, which will be exponential in the size of the original succinct game)

**Questions:**

Besides the review, it would be good to obtain some more understanding on the challenges one has to overcome to apply the "kernelization" trick in these settings? How much can one use from previous work as opposed to having to develop new ideas?

**Ethical Concerns:**

["NO or VERY MINOR ethics concerns only"]

**Final Justification:**

After engaging in the rebuttal: a strong paper with a clear case for acceptance.

**Limitations:**

Yes

**Quality:**

3

**Strengths And Weaknesses:**

Assessment:
-  I found the paper very well structured and written, and I enjoyed reading through it.
- The technical results are novel and focus on an area that has received increased attention.
- The main idea is to extend kernelization tricks from previous work. This is not immediate and requires additional work (e.g., second moments). However the idea itself mostly follows previous work, so the novelty is curtailed by that.
- There are no numerics. How does this approach compare “in practice” on standard network congestion games, eg Sioux Fall? In other words: is the analysis better, or is the algorithm itself better?
- I also note that in certain bandit / semi-bandit settings, the authors are not the first to obtain a dependance on the size of the game that is polynomial (e.g., compare with previous work cited where their dependence is polynomial, albeit with a bad exponent d^10). It would be good to make this clear in the intro too.

---

> ### Author Rebuttal · Authors · 2025-07-30
>
> Dear reviewer,
>
> Thank you for your time, and your positive evaluation! We reply to your remarks and questions below:
>
> > The main idea is to extend kernelization tricks from previous work. This is not immediate and requires additional work (e.g., second moments). However the idea itself mostly follows previous work, so the novelty is curtailed by that.
>
> We certainly do not claim any primarcy in introducing kernelization as a concept: the novelty of our contributions lie in extending it to settings with partial information (semi-bandit and bandit feedback) and the polyhedral structure of the underlying game (such as Colonel Blotto and Matroid Congestion Games).
>
> In particular, our main technical novelties are as follows:
>
> 1. Kernelization in the (semi-)bandit setting: We show how to use kernelization not only for first-moment calculations, as in prior work on full-information feedback [1], but also to efficiently compute second moments and perform exact sampling. These operations are essential in the (semi-)bandit setting and, to the best of our knowledge, had not been addressed in earlier work.
> 2. We utilize novel kernelization techniques to classes of polyhedral games that had not been analyzed from this perspective before:
>     - In Graphic Matroid Congestion Games, we design novel algorithms and analysis (see Appendix H) for exact sampling and moment computation, resolving a long-standing open problem posed by Cesa-Bianchi & Lugosi [3] on how to perform exact sampling from the ComBand algorithm (and thus MWU) in spanning trees. In particular, the kernels used for sampling are on the conditional polytope, that is on subsets of the original actions where some coordinates are restricted to have a specific 0-1 value. Computing such conditional kernels presents additional challenges compared to standard kernels: while standard kernels can be obtained by directly applying the Matrix Tree Theorem as a black box, conditional kernels require several non-trivial technical steps before the theorem can be applied.
>     - In Colonel Blotto games, we introduce an efficient action representation of size $nk$, develop a kernelization procedure in this representation, and apply acceleration techniques to reduce the amortized kernel computation time to $O(\log n)$. This kernelization also resolves an open question from [4]. Our techniques share similarities with those in [5] for the kernelization of m-sets, in the sense that, in both settings, a kernel can be computed via specific coefficients of some polynomial (see Eq. 3 for Colonel Blotto); however, designing fast kernelization algorithms in Colonel Blotto is substantially more challenging because the polynomial factors are $n$-degree polynomials as opposed to the binomials used for m-sets (see also Algorithm 4, Appendix G).
>
> ---
> > Besides the review, it would be good to obtain some more understanding on the challenges one has to overcome to apply the "kernelization" trick in these settings? How much can one use from previous work as opposed to having to develop new ideas?
>
> We used kernelization to efficiently implement some demanding operations that appear in our learning algorithms.
>
> In the semi-bandit case, we designed the learning algorithm from scratch. It turned out that the unbiased estimator uses the first moments of the policy and thus we could directly use the kernel trick proposed in [5] for the calculation of first moments in polyhedral games.
>
> An extra demanding operation required for the semi-bandit and the bandit learning is sampling from the player's policy. This operation is not required in the full information setting and we had to design from scratch a method for efficient sampling based on kernelization. In particular, the kernels used for sampling are on the conditional polytope, that is on subsets of the original actions where some coordinates are restricted to have a specific 0-1 value. The computation of such conditional kernels has extra challenges compared to the standard kernels as can be seen in our study of Graphic Matroid Congestion Games (Appendix H). In these polyhedra, the standard kernels could be computed by using the Matrix Tree Theorem as a blackbox but conditional kernels required several technicalities before the Matrix Tree Theorem could be applied.
>
> In the bandit setting, second moment calculation is required for the construction of the loss estimator. Inspired by the ideas of [5] for the use of kernels in first moment calculation we extend the idea to the second moment. In this case, the formulas are slightly more complicated and the use of inclusion-exclusion principle is required as an extra step. This is a significant difference compared to the first moment calculation and actually opens a path to generalizing the technique to higher order moments. However, note that the use of inclusion-exclusion principle for higher order moments has the limitation that the dependence on the degree of the moment is exponential. This is the reason that we can not directly use this trick for the calculation of conditional kernels required in sampling.
>
> Regarding applications of kernelization on specific polytopes of interest, we considered Colonel Blotto and Graphic Matroids, two games that have not been studied through the prism of kernelization before. In Graphic Matroids, we observed that kernel computation can be performed by a direct application of the Matrix Tree Theorem. However, in order to accelerate the calculation of groups of kernels we use some technical precomputing based on rank 1 modifications. For the calculation of conditional kernels used for sampling there are significant challenges as we described above. For an overview of our techniques for the kernelization of Colonel Blotto, see our answer to the previous question, in particular paragraph 2(b).
>
> ---
> > There are no numerics. How does this approach compare “in practice” on standard network congestion games, eg Sioux Fall? In other words: is the analysis better, or is the algorithm itself better?
>
> The way that OpenReview has been set up this year, it is not possible to include any simulations during the rebuttal stage (or even the result thereof through an anonymized picture or github link). However, we will be happy to take advantage of the extra page to include proof-of-concept experiments aimed at validating the efficiency of our approach—specifically, measuring $\mathrm{Reg}(T)/T$ as a function of $T$ until convergence to an ε-CCE.
>
> In more detail, we will add the following experimental setups:
>
> 1. Experiments on congestion games under (semi-)bandit feedback. We will use the Sioux Falls dataset (part of the well-known TransportationNetworks collection), which has been widely used in the literature to evaluate regret-like performance in congestion games. The dataset represents a network with 24 nodes and 76 edges. We will use 100 players and the standard linear cost function per edge.
> 2. Experiments on Colonel Blotto games under (semi-)bandit feedback. We will use different seeds to produce randomized Colonel Blotto games consisting of 10 players, 100 battlefields and 1000 soldiers per player. The reward function will be defined as "Winner Takes All" (that is, the battlefield reward of player $i$ is 1 if player $i$ assigned more soldiers in this battlefield than all other players, otherwise 0).
>
> ---
> > I also note that in certain bandit / semi-bandit settings, the authors are not the first to obtain a dependance on the size of the game that is polynomial (e.g., compare with previous work cited where their dependence is polynomial, albeit with a bad exponent d^10). It would be good to make this clear in the intro too.
>
> Indeed our paper is not the first to obtain a polynomial dependence on the game parameters $d$ and $m$. As we have discussed in the paper, for generic polyhedral games (that is, without considering a specific structure, e.g., Colonel Blotto), [1] and [2] proposed algorithms for continuous action spaces, which can be extended to polyhedral games achieving a regret bound of $\mathcal{O}(md^{7/2}\sqrt{T})$ and $\mathcal{O}(md^{2}\sqrt{T})$, respectively. However, the above bounds combined with a per-iteration complexity, which heavily depends on $d$, result in impractical runtime complexity results for learning CCE. In particular, the runtime of the algorithm in [1] to find CCE scales as $d^{10}$, while that of [2] scales as $d^9$; both exhibiting impractically large dependence on the game parameters. We have already referred to these previous works in the introduction, but we can use the extra space to make a more extended comparison.
>
> Regarding the applications of interest (Colonel Blotto and Congestion games), we have included a discussion in Appendix A (Further Related Work) with previous works appearing in the Tables. We could incorporate that discussion into the introduction as well.
>
> ---
> Thank you again for your time, input and positive evaluation—we remain at your disposal if you have any further questions!
>
> Kind regards,
>
> The authors
>
> ---
>
> ### References
>
> [1] Lee et al. "Bias no more: high-probability data-dependent regret bounds for adversarial bandits and mdps." NeurIPS (2020).
>
> [2] Zimmert and Lattimore. "Return of the bias: Almost minimax optimal high probability bounds for adversarial linear bandits." Conference on Learning Theory. 2022.
>
> [3] Cesa-Bianchi, Nicolo, and Gábor Lugosi. "Combinatorial bandits." Journal of Computer and System Sciences 78.5 (2012): 1404-1422.
>
> [4] Beaglehole, Daniel, et al. "Sampling equilibria: Fast no-regret learning in structured games." Proceedings of the 2023 Annual ACM-SIAM Symposium on Discrete Algorithms (SODA). Society for Industrial and Applied Mathematics, 2023.
>
> [5] Farina, Gabriele, et al. "Kernelized multiplicative weights for 0/1-polyhedral games: Bridging the gap between learning in extensive-form and normal-form games." International Conference on Machine Learning. PMLR, 2022.

---

> > ### Comment · Reviewer_wspx · 2025-08-04
> > **Thanks**
> >
> > Thanks for the comprehensive answer. On balance, I am positive on acceptance of the manuscript.
> >
> > The only point I would like to reiterate on is the need for some form of numerics in the final version of the manuscript. Please, make sure you do so, as per your previous comment above. Nice work!

---

> > > ### Author Response · Authors · 2025-08-05
> > >
> > > Thank you for your response and your positive evaluation. We will make sure to include the numerical analysis in the final version.
> > >
> > > Kind regards,
> > >
> > > The authors

---

### Official Review · Reviewer_pKuV · 2025-07-03

**Clarity:** 3
**Significance:** 3
**Originality:** 3
**Rating:** 4
**Confidence:** 1

**Summary:**

This paper investigates how to efficiently learn coarse correlated equilibria (CCE) in large structured games (polyhedral games) when players have only partial feedback—that is, they only observe the outcomes of the actions they took, rather than all possible actions. The key idea is to extend recent kernelization techniques that have previously worked only when full information is available to settings where only bandit or semi-bandit feedback is given.

The paper develops new algorithms for this setting, analyzes their performance in several game types (e.g., Colonel Blotto, congestion games), and claims significant improvements in both theoretical regret bounds and runtime. The paper appears to resolve some known open problems in this domain.

**Questions:**

1. Could the authors provide simple, concrete examples (e.g., toy games) to illustrate how the algorithms work in practice?

2. Are there any empirical results or experiments—even small simulations—that could demonstrate the effectiveness or practicality of the approach?

3. How broadly applicable are these techniques beyond the specific game classes discussed (e.g., Colonel Blotto, matroid games)?

4. Are there existing systems or domains where these methods could be deployed or tested?

5. What are the main barriers to implementation? Are the kernel computations feasible for moderate-scale problems?

**Ethical Concerns:**

["NO or VERY MINOR ethics concerns only"]

**Quality:**

3

**Strengths And Weaknesses:**

Strengths
1. The paper is clearly written, with helpful figures and detailed explanations that aid understanding despite the technical subject matter.

2. It appears to be a significant theoretical contribution, pushing the boundaries of what’s possible in a well-studied class of game-theoretic learning problems.

3. The organization is effective, with a logical flow from problem motivation to technical development and results.

4. The comparison tables with prior work are helpful for understanding claimed improvements.

Weaknesses
1. The paper is technically dense, and many results depend on domain-specific knowledge of combinatorial games, kernel methods, and no-regret learning, which may limit accessibility to a broader audience.

2. There are no empirical experiments or simulations to demonstrate the practical utility or performance of the proposed algorithms.

3. From a non-expert (in normal-form games and polyhedral games) perspective, it is difficult to judge the novelty or correctness of the technical contributions or to verify the extent to which the improvements over previous work are significant.

4. The assumptions and applicability to real-world systems are not always clear—for example, it is not obvious where such structured games arise in practice or how feasible these methods would be to implement.

---

> ### Author Rebuttal · Authors · 2025-07-30
>
> Dear reviewer,
>
> Thank you for your time, constructive input, and positive evaluation! We reply to your remarks and questions below:
>
> ---
> > The paper is technically dense, and many results depend on domain-specific knowledge of combinatorial games, kernel methods, and no-regret learning, which may limit accessibility to a broader audience.
>
> We tried to strike a balance between describing concepts that are relatively well known in the fields (such as no-regret learning), and providing the necessary intuition and background for our contributions. We understand however that some more specialized topics—such as kernelization and combinatorial games—could have been explained in more detail, and we will be happy to take advantage of the extra page available in the camera-ready round to do so. In the same spirit, we will also include a non-technical, high-level description of our results and the challenges involved for a broader audience—that is, not necessarily with a background in online learning and/or game theory.
>
> ---
> > From a non-expert (in normal-form games and polyhedral games) perspective, it is difficult to judge the novelty or correctness of the technical contributions or to verify the extent to which the improvements over previous work are significant.
>
> While our work builds on the general idea of kernelization from prior literature, our main novelty lies in extending it to previously unexplored settings—both in terms of feedback models (semi-bandit and bandit) and game structures (such as Colonel Blotto and Matroid Congestion Games).
>
> In particular, our main technical novelties are as follows:
>
> 1. Kernelization in the (semi-)bandit setting: We show how to use kernelization not only for first-moment calculations, as in prior work on full-information feedback [1], but also to efficiently compute second moments and perform exact sampling. These operations are essential in the (semi-)bandit setting and, to the best of our knowledge, had not been addressed in earlier work.
> 2. We employ new kernelization techniques for classes of polyhedral games that had not been analyzed from this perspective before:
>     - In Graphic Matroid Congestion Games, we design novel algorithms and analysis (see Appendix H) for exact sampling and moment computation, resolving a long-standing open problem posed by Cesa-Bianchi & Lugosi [7] on how to perform exact sampling from the ComBand algorithm (and thus MWU) in spanning trees. In particular, the kernels used for sampling are on the conditional polytope, that is on subsets of the original actions where some coordinates are restricted to have a specific 0-1 value. Computing such conditional kernels presents additional challenges compared to standard kernels: while standard kernels can be obtained by directly applying the Matrix Tree Theorem as a black box, conditional kernels require several non-trivial technical steps before the theorem can be applied.
>     - In Colonel Blotto games, we introduce an efficient action representation of size $nk$, develop a kernelization procedure in this representation, and apply acceleration techniques to reduce the amortized kernel computation time to $O(\log n)$. This kernelization also resolves an open question from [8]. Our techniques share similarities with those in [6] for the kernelization of m-sets, in the sense that, in both settings, a kernel can be computed via specific coefficients of some polynomial (see Eq. 3 for Colonel Blotto); however, designing fast kernelization algorithms in Colonel Blotto is substantially more challenging because the polynomial factors are $n$-degree polynomials as opposed to the binomials used for m-sets (see also Algorithm 4, Appendix G).
> 3. Our regret analysis paired with the above contributions led to a significant improvement of the total runtime to learn CCE in terms of the game parameters.
>
> We will make sure to highlight the above discussion in the camera-ready version of the paper.
>
> ---
> > There are no empirical experiments or simulations to demonstrate the practical utility or performance of the proposed algorithms. Could the authors provide simple, concrete examples (e.g., toy games) to illustrate how the algorithms work in practice? Are there any empirical results or experiments—even small simulations—that could demonstrate the effectiveness or practicality of the approach?
>
> The way that OpenReview has been set up this year, it is not possible to include any simulations during the rebuttal stage (or even the result thereof through an anonymized picture or github link). However, we will be happy to take advantage of the extra page to include proof-of-concept experiments aimed at validating the efficiency of our approach—specifically, measuring $\mathrm{Reg}(T)/T$ as a function of $T$ until convergence to an ε-CCE.
>
> In more detail, we will add the following experimental setups:
> 1. Experiments on congestion games under (semi-)bandit feedback. We will use the Sioux Falls dataset (part of the well-known TransportationNetworks collection), which has been widely used in the literature to evaluate regret-like performance in congestion games. The dataset represents a network with 24 nodes and 76 edges. We will use 100 players and the standard linear cost function per edge.
> 2. Experiments on Colonel Blotto games under (semi-)bandit feedback. We will use different seeds to produce randomized Colonel Blotto games consisting of 10 players, 100 battlefields and 1000 soldiers per player. The reward function will be defined as "Winner Takes All" (that is, the battlefield reward of player $i$ is 1 if player $i$ assigned more soldiers in this battlefield than all other players, otherwise 0).
>
> ---
> > How broadly applicable are these techniques beyond the specific game classes discussed (e.g., Colonel Blotto, matroid games)?
>
> We have considered some of the most prominent examples of polyhedral games and have demonstrated a wide array of kernelization techniques (i.e., generating functions in Colonel Blotto, weighted counting in Graphic Matroid Congestion Games and dynamic programming in Network Congestion Games). We believe that our techniques may be leveraged for other polyhedral settings as well, but we leave such a direction as an avenue for future research.
>
> ---
> > The assumptions and applicability to real-world systems are not always clear, for example, it is not obvious where such structured games arise in practice or how feasible these methods would be to implement. Are there existing systems or domains where these methods could be deployed or tested?
>
> All game settings we consider are of particular interest in the theoretical computer science and game theory community, as they can model many real-world applications. The Colonel Blotto games are often used to model political campaigns [1], where battles represent voting districts, and also have applications in security problems in IoT networks and cognitive radio [2] and ecological modeling [3]. Network congestion games have been widely used to model traffic congestion and routing problems [4], while graphic matroid congestion are used to model network design problems in which players compete for the edges of a graph in order to build a spanning tree [5].
>
> ---
> > What are the main barriers to implementation? Are the kernel computations feasible for moderate-scale problems?
>
> Yes, indeed, the overall kernel computation time required for an iteration of our method is $O(nk\log n)$ for the full-information and semi-bandit and $O(n^2 k^2)$ for the bandit setting in Colonel Blotto, $O(|E||V|^ω)$ for the full-information and semi-bandit and $O(|E||V|^{ω+1} + |E|^2|V|^2)$ for the bandit setting in graphic matroid congestion games (where $ω < 2.5$). For instance, consider the semi-bandit setting. Our method can be implemented efficiently for an input of $n=10^5$ and $k=10^4$ in Colonel Blotto, and for a complete graph of 100 nodes in graphic matroid congestion games.
>
> ---
> Thank you again for your input and positive evaluation—please do not hesitate to reach out if you have any more questions or remarks, we are looking forward to a further constructive exchanges during the discussion phase!
>
> Kind regards,
>
> The authors
>
> ---
>
>
> ### References
>
> [1] Behnezhad, Soheil, et al. "Optimal strategies of blotto games: Beyond convexity." Proceedings of the 2019 ACM Conference on Economics and Computation. 2019.
>
> [2] Min, Minghui, et al. "Defense against advanced persistent threats in dynamic cloud storage: A colonel blotto game approach." IEEE Internet of Things Journal 5.6 (2018): 4250-4261.
>
> [3] Golman, Russell, and Scott E. Page. "General Blotto: games of allocative strategic mismatch." Public Choice 138.3 (2009): 279-299.
>
> [4] Roughgarden, Tim, and Éva Tardos. "How bad is selfish routing?." Journal of the ACM (JACM) 49.2 (2002): 236-259.
>
> [5] Ackermann, Heiner, Heiko Röglin, and Berthold Vöcking. "On the impact of combinatorial structure on congestion games." Journal of the ACM (JACM) 55.6 (2008): 1-22.
>
> [6] Farina, Gabriele, et al. "Kernelized multiplicative weights for 0/1-polyhedral games: Bridging the gap between learning in extensive-form and normal-form games." International Conference on Machine Learning. PMLR, 2022.
>
> [7] Cesa-Bianchi, Nicolo, and Gábor Lugosi. "Combinatorial bandits." Journal of Computer and System Sciences 78.5 (2012): 1404-1422.
>
> [8] Beaglehole, Daniel, et al. "Sampling equilibria: Fast no-regret learning in structured games." Proceedings of the 2023 Annual ACM-SIAM Symposium on Discrete Algorithms (SODA). Society for Industrial and Applied Mathematics, 2023.

---

### Official Review · Reviewer_zzYs · 2025-07-20

**Clarity:** 3
**Significance:** 3
**Originality:** 3
**Rating:** 5
**Confidence:** 3

**Summary:**

This paper is concerned with learning coarse correlated equilibria (CCE) in normal form games with an exponentially large number of This paper is concerned with learning coarse correlated equilibria (CCE) in normal form games with an exponentially large number of actions per player, making the problem non-trivial and computationally expensive. While existing works have addressed the challenge via kernelization, they have only done so in the full-information setting; and the ones that solve the games under bandit-feedback have impractical runtimes. By building on the existing algorithm for bandit-feedback, namely the GeometricHedge, and kernelization framework, the authors propose a new computationally efficient framework that can learn CCE of several polyhedral games under partial-information settings. The authors provide efficient kernalization approach for three different games: the multi-player Colonel Blotto, graphic matroid and network congestion game.

**Questions:**

1. Kernelization is the core of the proposed method, and the authors provide kernelization techniques for each of the three games. My questions are:

   a) Do these three techniques encompass the majority of the polyhedral games?

   b) Do you envision a higher‑level framework into which any polyhedral game can be cast, so that a “universal” kernelization technique could be applied?

2. What do you think the challenges are when it comes to extending your method to solve for tighter equilibrium concepts: both CE and NE? Could you comment on the complexity of reaching an NE with the current approach?

**Ethical Concerns:**

["NO or VERY MINOR ethics concerns only"]

**Final Justification:**

I thank the authors for their thorough response. They have detailed the additional experiments they plan to include, rather than merely saying “we will add empirical results”, which directly addresses my primary concern about insufficient empirical evidence. Accordingly, I am raising my score to 5.

**Limitations:**

yes

**Quality:**

3

**Strengths And Weaknesses:**

**Strengths**
- The paper is well written with clearly laid out motivation for studying the problem of **efficiently** learning CCEs of polyhedral games under partial-information setting.
- Detailed pseudocode for the proposed algorithm and proofs.



**Limitations**
- While theoretical complexity are provided and proved, empirical studies are lacking. Given that one of the motivations for studying the problem was lack of "efficient implementation" of GeometricHedge algorithm, I think empirical results, such as: average regret over iterations, would make the paper stronger.

---

> ### Author Rebuttal · Authors · 2025-07-30
>
> Dear reviewer,
>
> Thank you for your time, and your positive evaluation! We reply to your remarks and questions below:
>
>
> > While theoretical complexity are provided and proved, empirical studies are lacking. Given that one of the motivations for studying the problem was lack of "efficient implementation" of GeometricHedge algorithm, I think empirical results, such as: average regret over iterations, would make the paper stronger.
>
> The way that OpenReview has been set up this year, it is not possible to include any simulations during the rebuttal stage (or even the result thereof through an anonymized picture or github link). However, we will be happy to take advantage of the extra page to include proof-of-concept experiments aimed at validating the efficiency of our approach—specifically, measuring $\mathrm{Reg}(T)/T$ as a function of $T$ until convergence to an ε-CCE.
>
> In more detail, we will add the following experimental setups:
> 1. Experiments on congestion games under (semi-)bandit feedback. We will use the Sioux Falls dataset (part of the well-known TransportationNetworks collection), which has been widely used in the literature to evaluate performance in congestion games. The dataset represents a network with 24 nodes and 76 edges. We will use 100 players and standard linear cost functions per edge.
> 2. Experiments on Colonel Blotto games under (semi-)bandit feedback. We will use different seeds to produce randomized Colonel Blotto games consisting of 10 players, 100 battlefields and 1000 soldiers per player. The reward function will be defined as "Winner Takes All" (that is, the battlefield reward of player $i$ is 1 if player $i$ assigned more soldiers in this battlefield than all other players, otherwise 0).
>
> ---
> > Do these three techniques encompass the majority of the polyhedral games?
>
> We have considered some of the most prominent examples of polyhedral games and have demonstrated a wide array of kernelization techniques (i.e., generating functions in Colonel Blotto, weighted counting in Graphic Matroid Congestion Games and dynamic programming in Network Congestion Games). We believe that our techniques may be leveraged for other polyhedral game settings as well, but we leave such a direction as an avenue for future research.
>
> ---
> > Do you envision a higher‑level framework into which any polyhedral game can be cast, so that a “universal” kernelization technique could be applied?
>
> Both yes and no. The proposed learning algorithms can indeed be applied to any polyhedral game as a general, umbrella framework and achieve the same no-regret guarantees (in terms of $T$, $m$, and $d$). The obstacle however is the kernelization part, which is not application-agnostic and indeed needs to leverage the specific structure of the class of polyhedral games in question (in terms of the facet structure of the underlying polyhedra etc.)
>
> We should note here that not all polyhedral structures are amenable to efficient kernelization, since for some classes of polyhedra kernelization is provably hard. For example, consider the problem of finding Hamiltonian cycles in a graph. In this setting, the action vectors are binary edge-incidence vectors of size $|E|$, similarly to the case of congestion games that we consider in the paper. However, in this case, kernelization is NP-hard, because the all-ones kernel $K(1,1)$ is equal to the total number of Hamilton cycles in the graph and if positive it would imply the existence of a Hamiltonian cycle, and vice versa (thus sovling a well-known NP-hard decision problem).
>
> ---
> > What do you think the challenges are when it comes to extending your method to solve for tighter equilibrium concepts: both CE and NE? Could you comment on the complexity of reaching an NE with the current approach?
>
> - **For Nash equilibria:** there is no connection to no-regret learning in the general case and thus kernelization does not seem to be relevant. Note that finding approximate Nash equilibria in general-sum normal-form games is PPAD-hard, and normal-form games are a special case of polyhedral games where the action polyhedron is a simplex. Thus, even through kernelization there is no hope for efficient learning of Nash equilibria in general. In specific cases however, kernelization can indeed be used to learn Nash equilibria efficiently:
>     1. First, in zero-sum games, due to the so-called equilibrium collapse property, the set of coarse correlated equilibria coincides in value with the set of Nash equilibria in zero-sum games and  the marginalization of a CCE is a Nash equilibrium in these games. Thus, our kernelization method for learning CCE can be used to learn Nash as well.
>     2. Second, in potential games (such as the congestion games we consider), even though there is no collapse, Nash equilibria can be easily learnt, and there exist algorithms based on MWU to learn Nash equilibria in the full information setting (see [1]). Since kernelization is used to implement MWU efficiently, the approach of [1] paired with the kernelization trick can be used to efficiently learn Nash equilibria in polyhedral potential games.
>
> - **For correlated equilibria:** it is a major open question in the field whether one can design FPTAS algorithms to find CE in polyhedral games. Only a PTAS algorithm is known [2,3], which is a learning approach and depends on efficient MWU implementation in the game's action space. Thus, kernelization can be also used in this framework and yield efficient (PTAS) algorithms for learning CE in polyhedral games.
>
> ---
> Thank you again for your constructive input and positive evaluation—please do not hesitate to reach out if you have any further questions or remarks, we are looking forward to a constructive exchange during the rebuttal phase!
>
> Kind regards,
>
> The authors
>
> ---
> ### References
>
> [1] Cen, Shicong, Fan Chen, and Yuejie Chi. "Independent natural policy gradient methods for potential games: Finite-time global convergence with entropy regularization." 2022 IEEE 61st Conference on Decision and Control (CDC). IEEE, 2022.
>
> [2] Dagan, Yuval, et al. "From external to swap regret 2.0: An efficient reduction for large action spaces." Proceedings of the 56th Annual ACM Symposium on Theory of Computing. 2024.
>
> [3] Peng, Binghui, and Aviad Rubinstein. "Fast swap regret minimization and applications to approximate correlated equilibria." Proceedings of the 56th Annual ACM Symposium on Theory of Computing. 2024.

---

> ### Comment · Reviewer_zzYs · 2025-08-03
> **Thank you for the response**
>
> Thank you for addressing my questions and for the detailed responses. I would like to commend the authors for outlining the detailed proof-of-concept experiments to show the convergence of average regret. I believe these results would make the paper complete.

---

> > ### Author Response · Authors · 2025-08-05
> >
> > Thank you for your response and your positive feedback. We will make sure to include the experiments in the final version.
> >
> > Kind regards,
> >
> > The authors

---

### Decision · Program_Chairs · 2025-09-17

**Decision:**

Accept (poster)

**Comment:**

This paper develops efficient payoff-based learning algorithms for computing coarse correlated equilibria (CCE) in polyhedral games, which have exponentially large action spaces and combinatorial structure (including Colonel Blotto and congestion games). Building upon a recent development that enables to “kernelize” no-regret like dynamics, the authors are able to provide algorithms that provably converge to a CCE with running time that depends polynomially on the size of the succinct game.

The reviewers all agree that this submission makes progress on fundamental and well-studied challenges in (algorithmic) game theory and online learning. The contribution is technically interesting, and the submission is written very clearly. The only weakness mentioned by multiple reviewers are missing experimental results. The authors have, however, conducted experiments in the discussion phase and provided a summary of their findings. They promised to include this in the final version of the paper. The reviewers were all satisfied with this and consider also the experimental results as promising. Overall, the clear suggestion is to accept this submission.